# A Review of Image Super-Resolution Approaches Based on Deep Learning and Applications in Remote Sensing

Xuan Wang [1,†], Jinglei Yi [1,†], Jian Guo [1], Yongchao Song [1], Jun Lyu [1], Jindong Xu [1], Weiqing Yan [1], Jindong Zhao [1], Qing Cai [2,3] and Haigen Min [4,5,*]

1 School of Computer and Control Engineering, Yantai University, Yantai 264005, China
2 School of Data Science, The Chinese University of Hong Kong, Shenzhen 518172, China
3 School of Information Science and Technology, University of Science and Technology of China, Hefei 230026, China
4 School of Information Engineering, Chang'an University, Xi'an 710064, China
5 The Joint Laboratory for Internet of Vehicles, Ministry of Education-China Mobile Communications Corporation, Xi'an 710064, China
* Correspondence: hgmin@chd.edu.cn
† These authors contributed equally to this work.

**Abstract:** At present, with the advance of satellite image processing technology, remote sensing images are becoming more widely used in real scenes. However, due to the limitations of current remote sensing imaging technology and the influence of the external environment, the resolution of remote sensing images often struggles to meet application requirements. In order to obtain high-resolution remote sensing images, image super-resolution methods are gradually being applied to the recovery and reconstruction of remote sensing images. The use of image super-resolution methods can overcome the current limitations of remote sensing image acquisition systems and acquisition environments, solving the problems of poor-quality remote sensing images, blurred regions of interest, and the requirement for high-efficiency image reconstruction, a research topic that is of significant relevance to image processing. In recent years, there has been tremendous progress made in image super-resolution methods, driven by the continuous development of deep learning algorithms. In this paper, we provide a comprehensive overview and analysis of deep-learning-based image super-resolution methods. Specifically, we first introduce the research background and details of image super-resolution techniques. Second, we present some important works on remote sensing image super-resolution, such as training and testing datasets, image quality and model performance evaluation methods, model design principles, related applications, etc. Finally, we point out some existing problems and future directions in the field of remote sensing image super-resolution.

**Keywords:** image super-resolution; deep learning; remote sensing; model design; evaluation methods

## 1. Introduction

Agriculture, meteorology, geography, the military, and other fields have benefited from remote sensing imaging technology. In application scenarios such as pest and disease monitoring, climate change prediction, geological survey, and military target identification, remote sensing images are indispensable. Therefore, in order to realize remote sensing image applications and analyses, high-resolution remote sensing images are essential. Despite this, factors such as sensor noise, optical distortion, and environmental interference can adversely affect the quality of remote sensing images and make it difficult to acquire high-resolution remote sensing images. Image super-resolution aims to reconstruct high-resolution (HR) images from low-resolution (LR) images (as shown in Figure 1), which is a typical computer vision task to mitigate the effects of acquisition equipment and environmental factors on remote sensing imaging results and improve the resolution of remote sensing images. However, the SR problem assumes that low-pass-filtered (blurred)

LR data are a downsampled and noisy version of HR data. The loss of high-frequency information during the irreversible low-pass filtering and secondary sampling operations causes SR to be an ill-posed problem. In addition, the super-resolution (SR) operation is a pair of multiple mappings from LR to HR space, which can have multiple solution spaces for any LR input, so it is essential to determine the correct solution from it. Many methods have been proposed to solve such an inverse problem, which can be broadly classified into early interpolation-based methods [1–3], reconstruction-based methods [4–6], and learning-based methods [7–14]. Since interpolation-based methods, such as the bicubic interpolation method [15], typically upsample LR images to obtain HR images, although they are simple and fast, some high-frequency information is destroyed in the process, leading to a decrease in model accuracy. The reconstruction-based methods are implemented based on adding the prior knowledge of the image as a constraint to the super-resolution reconstruction process of the image. Based on the SoftCuts metric, [16] proposes an adaptive SR technique based on prior edge smoothing. Although this overcomes the problem of uncomfortable image super-resolution, it also has the disadvantages of slow convergence speed and high computational cost. To achieve super-resolution reconstruction, the learning-based method relies on a large number of LR and HR images as a priori information. In [17], local feature blocks are learned between LR and HR images using the neighbor embedding method. Nonetheless, if learning becomes difficult (for example, when super-resolution magnification damages detailed features in the image), the learning-based method will perform less well. Therefore, the currently popular super-resolution is based on deep learning, which learns the mapping between LR and HR image spaces to predict the missing high-frequency information in low-resolution images in a time-saving and efficient manner.

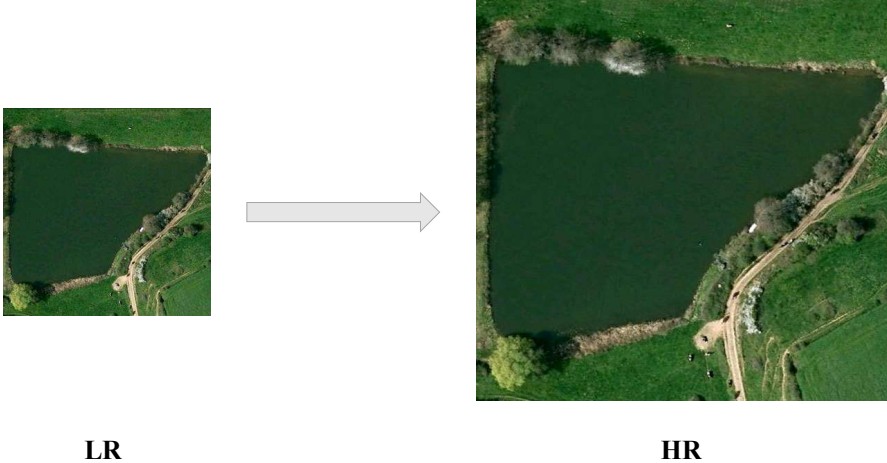

**LR**                                    **HR**

**Figure 1.** SR aims to reconstruct a high-resolution (HR) image from its degraded low-resolution (LR) counterpart.

The field of deep learning is continually developing. In recent years, many SR models based on deep learning have been proposed and have achieved significant results on benchmark test datasets of SR. Furthermore, the application of SR models to super-resolution tasks on remote sensing images has become an increasingly popular topic in the field of SR. Many attempts have been made by researchers to improve the performance of SR models on remote sensing images. In particular, Dong et al. first designed a model with three CNN layers, i.e., SRCNN [18]. Subsequently, Kim et al. increased the network depth to 20 in DRCN [19], and the experimental results were significantly improved compared with those of SRCNN [18]. On this basis, Liebel et al. [20] retrained SRCNN [18] using remote sensing image datasets to adapt the model to the multispectral nature of remote sensing data. VDSR [21] introduced residual learning and gradient cropping while increasing the number of network layers and solved the problem of processing multi-scale images

in a single framework. LGCnet [22] is a combined local–global network based on VDSR proposed by Lei et al. The problem of missing local details in remote sensing images is solved by combining shallow and deep features through a branching structure, which makes full use of both local and global information. To solve the discomfort problem of the super-resolution of images, Guo et al. developed a dual regression model, DRN [23]. This model learns mappings directly from LR images without relying on HR images. Overall, in order to achieve better results, most SR methods are improved in terms of the following aspects: network architecture design, selection of the loss function, development of the learning strategy, etc.

Due to their superior performance, the exploration of deep-learning-based SR methods is growing deeper. Several survey articles on SR have been published. However, most of these reports highlight various evaluation metrics for the reconstruction results of SR algorithms. In this paper, instead of simply summarizing available survey works [24–28], we provided a comprehensive overview of SR methods, focusing on the principles and processes of deep learning to demonstrate their performance, innovation, strength and weakness, relevance, and challenges, while focusing on their application to remote sensing images. Figure 2 shows the hierarchically structured classification of SR in this paper.

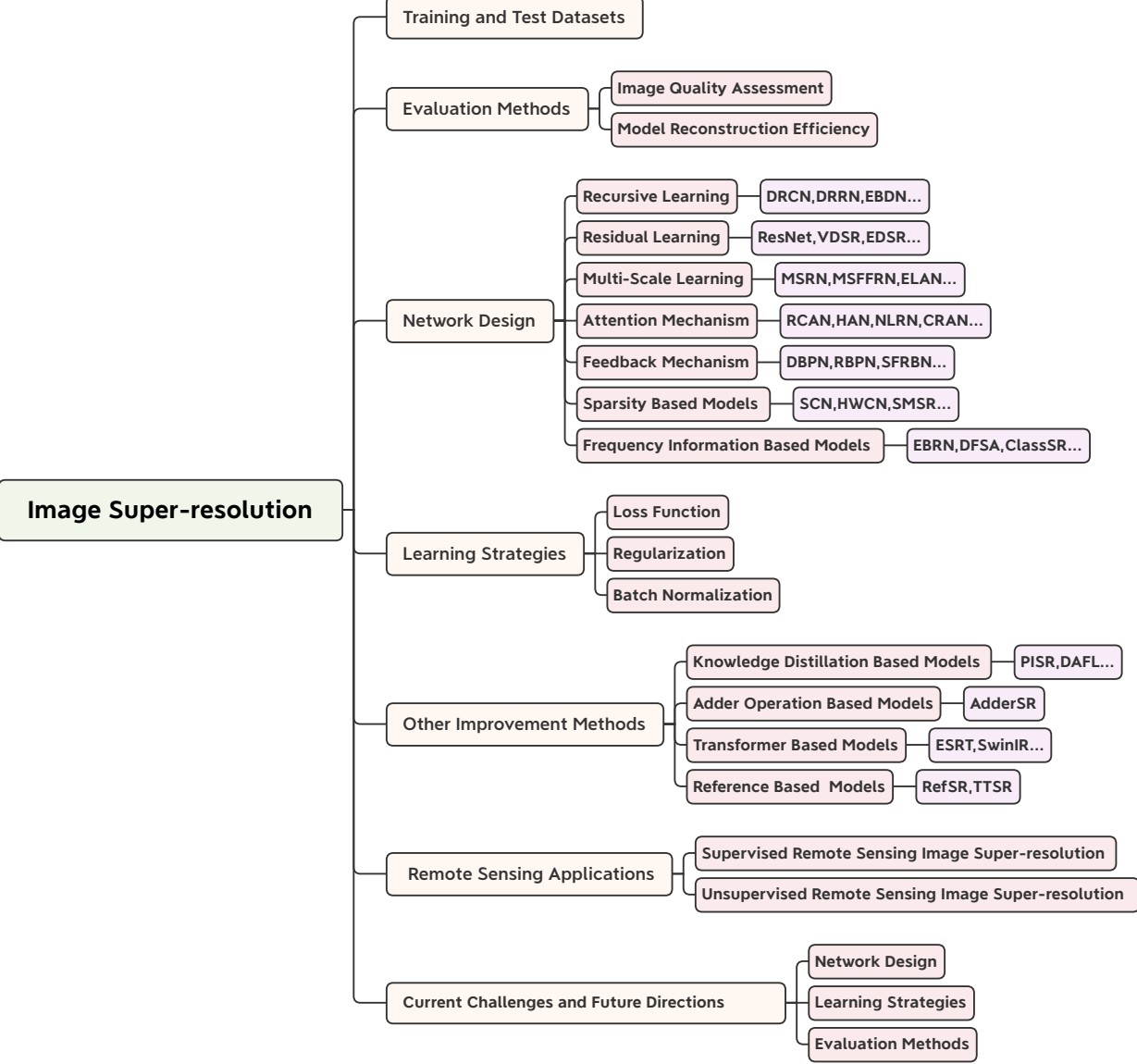

**Figure 2.** Hierarchically structured classification of SR in this paper.

The main contributions of this paper are as follows:

- We provide a comprehensive introduction to the deep-learning-based super-resolution process, including problem definitions, datasets, learning strategies, and evaluation methods, to give this review a detailed background.
- We classify the SR algorithms according to their design principles. In addition, we analyze the effectiveness of several performance metrics of representative SR algorithms on benchmark datasets, and some remote sensing image super-resolution methods proposed in recent years are also introduced. The visual effects of classical SR methods on remote sensing images are shown and discussed.
- We analyze the current issues and challenges of super-resolution remote sensing images from multiple perspectives and present valuable suggestions, in addition to clarifying future trends and directions for development.

The remaining sections of this review are arranged as follows. In Section 2, we briefly discuss what deep-learning-based SR is, the commonly used datasets, and the evaluation metrics. Section 3 describes in detail representative deep neural network architectures for SR tasks. In Section 4, several evaluation metrics are used to compare the performance of the SR methods mentioned in Section 3 and their application to remote sensing images. The applications of SR in remote domains are presented in Section 5. In Section 6, we discuss the current challenges and potential directions of SR. Finally, the work is concluded in Section 7.

## 2. Background

### 2.1. Deep-Learning-Based Super-Resolution

With advances in computing power, deep learning [29] in super-resolution has developed rapidly in recent years. Deep learning is a concept developed based on artificial neural networks [30], which is an extension of machine learning. Artificial neural networks imitate the way the human brain thinks, with artificial neurons as the computational units; the artificial neural network structure reflects the way these neurons are connected. The objective of deep learning is to determine the feature distribution of data by learning a hierarchical representation [31] of the underlying features. Specifically, deep learning continuously optimizes the super-resolution algorithm process through a series of learning strategies, such as deep network architecture, optimizer, and loss function design, while alleviating the ill-posed problem of super-resolution. Typically, the LR image $I_x$ is modeled as the output of the following degradation:

$$I_x = \left( I_y \otimes k \right) \downarrow_s + n, \tag{1}$$

where $I_y \otimes k$ denotes the convolution operation between the HR image $I_y$ and the degenerate blur kernel $k$ (e.g., double cubic blur kernel, Gaussian blur kernel, etc.), $\downarrow_s$ is the downsampling operation with scale factor $s$, and $n$ is the usually additive Gaussian white noise.

Deep learning differs from traditional algorithms because it can transfer the above processes into an end-to-end framework, saving time and efficiency. This is represented by the network structure of SRCNN [18], as shown in Figure 3. The image super-resolution process is roughly divided into three steps: feature extraction and representation, non-linear mapping, and image reconstruction. Specifically, first, the feature blocks are extracted from the low-resolution image by $9 \times 9$ convolution, and each feature block is represented as a high-dimensional vector. Then, each high-dimensional vector is non-linearly mapped to another high-dimensional vector by $5 \times 5$ convolution, where each mapped vector is a high-resolution patch. Finally, the final high-resolution image is generated by aggregating the above high-resolution patches by $5 \times 5$ convolution.

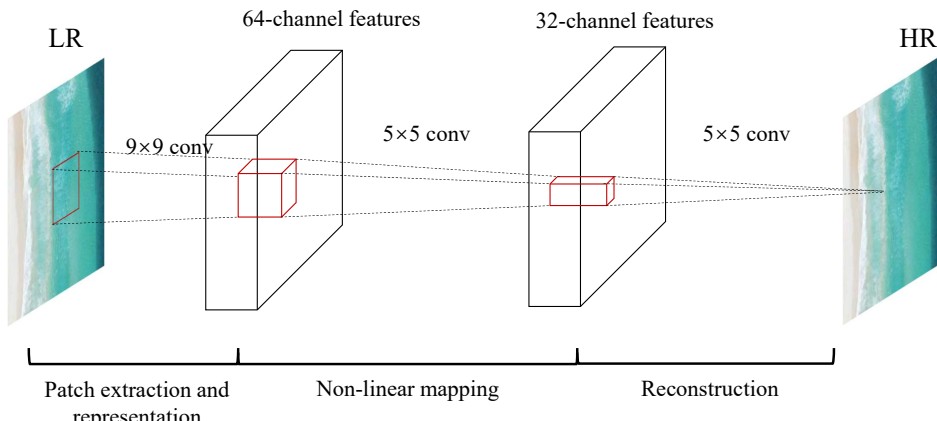

**Figure 3.** The network structure of SRCNN [18].

In comparison with natural images, remote sensing images differ in that (1) remote sensing images are captured from a distance of several hundred kilometers from the surface of the earth and are usually acquired by the use of aerial photography, land and ocean satellites, etc.; (2) most of the scenes in remote sensing images are forests, rivers, industrial areas, and airports, etc., which are typically large in scope, contain small objects, and have varied distribution forms; and (3) remote sensing images acquired under different weather conditions differ as well, due to factors such as lighting conditions on sensors, and clouds and fog that obscure the ground. The reconstruction of super-resolution remote sensing images, therefore, requires special considerations. For remote sensing images collected from forests and grasslands, the colors of the objects in the scene are very similar. It is difficult to classify the scene contents by color features alone. By referring to the texture features of these images, the "rough" forest and the "smooth" grass can be easily distinguished by the super-resolution method for these images.

### 2.2. Training and Test Datasets

Deep learning is a method of learning from data, and the goodness of the data plays an important role in the learning outcome of the model, with high-quality data being able to facilitate the improvement of the reconstruction performance of the deep learning SR-based model. Many diverse datasets for training and testing SR tasks have previously been proposed. Datasets commonly used for SR model training include BSDS300 [32], BSDS500 [33], DIV2K [34], etc. Similarly, BSD100 [32], Set5 [35], Set14 [36], Urban100 [37], etc. can be used to effectively test model performance. In particular, remote sensing image datasets such as AID [38], RSSCN7 [39], and WHU-RS19 [40] have been widely used in remote sensing image super-resolution tasks. In Table 1, we list some datasets that are commonly used in SR tasks (including the super-resolution of remote sensing images) and specify their image counts, image formats, resolutions, and content descriptions. Among them, the representative training dataset is the DIV2K [34] dataset, on which most SR models are trained. The DIV2K [34] dataset has three components: 800 training images, 100 validation images, and 100 test images. Set5 and Set14 are the classic test datasets for SR tasks, and they can accurately reflect the model performance. The OutdoorScene [41] dataset contains plants, animals, landscapes, reservoirs, etc., in outdoor scenes. AID [38] was originally used for the object detection task of remote sensing images, which contains 10,000 remote sensing images of $600 \times 600$ pixels, with scenes including airports, beaches, deserts, etc. RSSCN7 [39] contains 2800 remote sensing images from different seasons, arranged at four different scales, showing scenes such as farmland, parking lots, residential areas, and industrial areas. The WHU-RS19 [40] dataset comprises remote sensing images from 19 scenes, of which 50 images are included in each category. UC Merced [42] contains remote sensing images of 21 categories of scenes, 100 images per category, and each image size is $256 \times 256$ pixels. NWHU-RESISC45 [43] is published by Northwestern Polytechnic

University. The images represent 45 different categories of scenes, with 700 images per scene. RSC11 [44] is derived from Google Earth and contains 11 categories of scene, with each category having 100 images.

**Table 1.** Common datasets for image super-resolution (SR) and some remote sensing image datasets.

| Dataset | Amount | Resolution | Format | Short Description |
| --- | --- | --- | --- | --- |
| BSD300 [32] | 300 | (435, 367) | JPG | animal, scenery, decoration, plant, etc. |
| BSD500 [33] | 500 | (432, 370) | JPG | animal, scenery, decoration, plant, etc. |
| DIV2K [34] | 1000 | (1972, 1437) | PNG | people, scenery, animal, decoration, etc. |
| Set5 [35] | 5 | (313, 336) | PNG | baby, butterfly, bird, head, woman |
| Set14 [36] | 14 | (492, 446) | PNG | baboon, bridge, coastguard, foreman, etc. |
| T91 [45] | 91 | (264, 204) | PNG | flower, face, fruit, people, etc. |
| BSD100 [32] | 100 | (481,321) | JPG | animal, scenery, decoration, plant, etc. |
| Urban100 [37] | 100 | (984, 797) | PNG | construction, architecture, scenery, etc. |
| Manga109 [46] | 109 | (826, 1169) | PNG | comics |
| PIRM [47] | 200 | (617, 482) | PNG | animal, people, scenery, decoration, etc. |
| City100 [48] | 100 | (840,600) | RAW | city scene |
| OutdoorScene [41] | 10624 | (553, 440) | PNG | scenes outside |
| AID [38] | 10000 | (600, 600) | JPG | airport, bare land, beach, desert, etc. |
| RSSCN7 [39] | 2800 | (400, 400) | JPG | farmlands, parking lots, residential areas, lakes etc. |
| WHU-RS19 [40] | 1005 | (600, 600) | TIF | bridge, forest, pond, port, etc. |
| UC Merced [42] | 2100 | (256, 256) | PNG | farmland, bushes, highways, overpasses, etc. |
| NWHU-RESISC45 [43] | 31,500 | (256, 256) | PNG | airports, basketball courts, residential areas, ports, etc. |
| RSC11 [44] | 1232 | (512, 512) | TIF | grasslands, overpasses, roads, residential areas, etc. |

In addition to the datasets introduced above, datasets such as ImageNet [49], VOC2012 [50], and CelebA [51] for other image processing tasks were also introduced into the SR task.

### 2.3. Evaluation Methods

The evaluation index of image reconstruction quality can reflect the reconstruction accuracy of an SR model. Meanwhile, the number of parameters, running time, and computation of a model reflect the performance of an SR model. In this section, the evaluation methods of image reconstruction quality and reconstruction efficiency are introduced.

### 2.3.1. Image Quality Assessment

Due to the widespread use of image super-resolution techniques, evaluating the quality of reconstructed images has become increasingly important. Image quality refers to the visual properties of an image, and the methods of image quality evaluation, distinguished from the point of view of human involvement, include two branches: subjective and objective evaluation. Using subjective evaluation, we can determine the quality of an image (whether it appears realistic or natural) based on statistical analysis and with a human being as the observer. This type of method can truly reflect human perception. The objective evaluation of an organization is usually conducted based on numerical calculations utilizing some mathematical algorithm that can automatically calculate the results. In general, the former is a straightforward approach and more relevant to practical needs, but these methods are difficult to implement and inefficient. Therefore, objective evaluation methods are more widely used in image quality assessment, especially complete reference methods, and several commonly used methods for image quality assessment are described below.

### Peak Signal-to-Noise Ratio (PSNR)

PSNR [52] is one of the most popular objective image evaluation metrics in SR. Given a ground truth image $I_y$ with $N$ pixels and a reconstructed image $I_{SR}$, the PSNR can be defined by using the mean square error (MSE) as

$$PSNR = 10 \cdot \log_{10}\left(\frac{L^2}{MSE}\right), \tag{2}$$

$$MSE = \frac{1}{N}\sum_{i=1}^{N}(I_y - I_{SR})^2, \tag{3}$$

where $L$ refers to the peak signal, i.e., $L = 255$ for an 8-bit grayscale image. Although PSNR is relatively simple in its computational form and has a clear physical meaning, it essentially does not introduce human visual system characteristics into the image quality evaluation because it only considers MSE at the pixel level. Only the differences are analyzed purely from a mathematical perspective, resulting in the inability of PNSR to capture the differences in visual perception. However, it is more important to evaluate the constructive quality of the reconstructed image, so PNSR remains an accepted evaluation metric.

Structural Similarity (SSIM)

SSIM [52] is another popular image evaluation metric in the SR field. Unlike PSNR, which measures absolute error, SSIM belongs to the perceptual model and can measure the degree of distortion of a picture, as well as the degree of similarity between two pictures. As a full-reference objective image evaluation metric, SSIM is more in line with the intuition of the human eye. Specifically, SSIM is a comprehensive measure of similarity between images from three aspects, including structure, brightness, and contrast, which is defined as

$$SSIM = \left(l(I_{SR}, I_y)^\alpha \cdot c(I_{SR}, I_y)^\beta \cdot s(I_{SR}, I_y)^\gamma\right), \tag{4}$$

$$l(I_{SR}, I_y) = \frac{\left(2\mu_{I_{SR}}\mu_{I_y} + C_1\right)}{\mu_{I_{SR}}^2 + \mu_{I_y+}^2 C_1}, \tag{5}$$

$$c(I_{SR}, I_y) = \frac{\left(2\sigma_{I_{SR}}\sigma_{I_y} + C_2\right)}{\sigma_{I_{SR}}^2 + \sigma_{I_y}^2 C_2}, \tag{6}$$

$$s(I_{SR}, I_y) = \frac{\left(\sigma_{I_{SR}I_y} + C_3\right)}{\sigma_{I_{SR}}\sigma_{I_y} + C_3}, \tag{7}$$

where $C_1$, $C_2$, and $C_3$ are constants and $\alpha$, $\beta$, and $\gamma$ are weighting parameters. In order to avoid the case that the denominator is 0, $C_1 = (k_1 L)^2$, $C_2 = (k_2 L)^2$, $C_3 = \frac{C_2}{2}$, and $k_1 \ll 1, k_2 \ll 1$. SSIM takes values in the range of [0,1], and the larger the value, the higher the similarity between two images. MS-SSIM is a variant of SSIM, and due to the multivariate observation conditions, it takes into account the similarity between images at different scales and makes the image evaluation more flexible.

Mean Opinion Score (MOS)

MOS is a subjective evaluation method, usually using the two-stimulus method [53]. An observer directly rates the perception of image quality, and this result is mapped to a numerical value and averaged over all ratings, i.e., MOS. Many personal factors, such as emotion, professional background, motivation, etc., can impact the evaluation results when the observer performs the evaluation, which will cause the evaluation results to become unstable and not accurate enough to ensure fairness. Moreover, MOS is a time-consuming and expensive evaluation method because it requires the participation of the observer.

In addition to the above image evaluation metrics, there are many other image evaluation methods [54], such as the Natural Image Quality Evaluator (NIQE) [55], which is an entirely blind metric that does not rely on human opinion scores and expects a priori information to extract "quality-aware" features from images to predict their quality. The algorithmic process of NIQE is more accessible to implement than MOS. Learned Perceptual Image Patch Similarity (LPIPS) [56] is also known as "perceptual loss". Specifically, when evaluating the quality of super-resolution reconstructed images, it pays more attention to

the depth features of the images and learns the inverse mapping from the reconstructed images to a high resolution, and then calculates the L2 distance between them. Compared with the traditional PNSR and SSIM methods, LPIPS is more in line with human perception.

### 2.3.2. Model Reconstruction Efficiency
Storage Efficiency (Params)

When evaluating an SR model, the quality of the reconstructed images it outputs is essential. Still, the complexity and performance of the model need to be paid attention to as well in order to promote the development and application of image super-resolution in other fields while considering the output results of the losing model. The number of parameters, running time, and computational efficiency of an SR model are important indicators reflecting the efficiency of model reconstruction.

Execution Time

The running time of a model is a direct reflection of its computational power. The current popular lightweight networks not only have a relatively low number of parameters but also have short running times. If an SR model adds complex operations, such as attention mechanisms, this can lead to an increase in running time and affect the performance of the model. Therefore, the running time is also an essential factor in determining the performance of the model.

Computational Efficiency (Mult & Adds)

Since the algorithmic process in convolutional neural networks is primarily dependent upon multiplication and addition operations, the multiplicative addition operands are used to assess the computational volume of the model, as well as to indirectly reflect its computational efficiency. The size of the model and the running time are the influencing factors of the multiplicative–additive operands.

To conclude, when evaluating the SR model, it is also important to take into account the complexity and performance of the model.

## 3. Deep Architectures for Super-Resolution
### 3.1. Network Design

Network design is a key part of the deep learning process, and this section will introduce and analyze some mainstream design principles and network models in the super-resolution domain, as well as explain and illustrate some deep learning strategies. Finally, some design methods that deserve further exploration will be discussed.

### 3.1.1. Recursive Learning

Increasing the depth and width of the model is a common means to improve the performance of the network, but this brings with it a large number of computational parameters, as shown in Figure 4. Recursive learning is proposed to control the number of model parameters and to achieve the sharing of parameters in recursive modules. In simple terms, recursive learning means reusing the same module multiple times. DRCN [19] applies recursive learning to super-resolution problems by using a single convolutional layer as the recursive unit and setting 16 recursions to increase the perceptual field to $41 \times 41$ without introducing too many parameters. However, the superimposed use of recursive modules also poses some problems: gradient explosion or disappearance. Therefore, in DRRN [57], global and local residual learning is introduced to solve the gradient problem, i.e., ResBlock is used as the recursive unit to reduce the training difficulty. Ahn et al. [58] made improvements to the recursive application of ResBlock. They proposed a global and local cascade connection structure to further speed up the network training and make the transfer of information more efficient. In addition, the EBRN presented by Qiu et al. [59] uses recursive learning to achieve the differentiation of information with a different frequency, i.e., low-frequency information is processed with shallow modules in

the network, and high-frequency information is processed with deep modules. Recursive learning has also been widely used in some recent studies [60–62]. For example, in the SRRFN proposed by Li et al. [60], the recursive fractal module consists of a series of fractal modules with shared weights, which enables the reuse of model parameters.

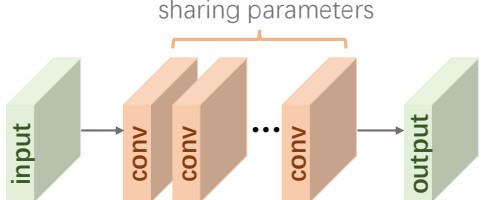

**Figure 4.** The structure of recursive learning.

### 3.1.2. Residual Learning

While recursive learning enables models to achieve a higher performance with as few parameters as possible, it also introduces the problem of exploding or vanishing gradients, and residual learning is a more popular approach to alleviate these problems. He et al. [63] proposed the use of residual learning in ResNet. It aims to mitigate the problem of exploding or disappearing gradients by constructing constant mappings using layer-hopping connections so that gradients in back-propagation can be passed directly to the network front-end through shortcuts, as shown in Figure 5. In image super-resolution tasks, low-resolution input images and high-resolution reconstructed images have most of the relevant information in terms of features, so only the residuals between them need to be learned to recover the lost information. In such a context, many residual learning based models [58,64–66] were proposed. Kim et al. proposed a profound super-resolution residual network VDSR [21] based on VGG-16, which has 20 layers, and takes the low-resolution image with bi-trivial interpolation as the input image. The residual information learned by the network is summed with the original input image as the output. Generally, the composition of the residual branch includes $3 \times 3$ convolutional layers, BN layers, and the relu activation function; some other ways this can be set up are mentioned in [67]. However, it is mentioned in EDSR [68] that the BN layer in the residual module is not suitable for super-resolution tasks because the distribution of colors of any image is normalized after BN layer processing. The original contrast information of the image is destroyed, which affects the quality of the output image of the network. Therefore, the BN layer is often chosen to be removed when designing residual modules in super-resolution tasks. RDN [64] proposes a residual dense block (RDB) that can fully preserve the features of the output of each convolutional layer. Nowadays, residual learning is a common strategy for super-resolution network design and has been applied in many models [69–73].

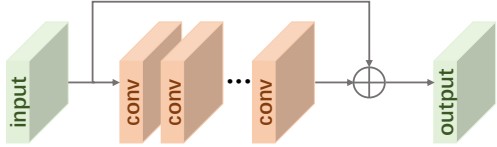

**Figure 5.** The structure of residual learning.

Although this global residual learning strategy achieves good results, global residual learning refers to the jump connection established from the input to the output. As the network levels deepen, global residual learning alone cannot recover a large amount of lost information, so researchers propose local residual learning, which is located in every few stacked layers and helps to preserve image details. A combination of global and local residual learning is applied in models such as VDSR [21], and EDSR [68].

### 3.1.3. Multi-Scale Learning

It has been pointed out [74,75] that images at different scales possess different features, and these rich features will help to generate high-quality reconstructed images. Therefore, multi-scale learning has been proposed to enable models to fully utilize features at different scales, while being applied to many SR models [76]. Li et al. [65] concluded that previous models were less robust to scale and less scalable, so multi-scale learning was applied to the SR task. He proposed a multi-scale residual module (MSRB) that used a $1 \times 1$ convolution kernel combined with $3 \times 3$ and $5 \times 5$ convolution kernels to obtain information at different scales (as shown in Figure 6), while local residual learning further improves the network training efficiency. In [77], the authors combined residual learning with multi-scale learning and proposed a multi-scale feature fusion residual block (MSFFRB) to extract and fuse image features of different scales. The multi-scale feature extraction and attention module (MSFEAAB) in [78] used convolutional kernels containing different sizes within the same layer to extract information of different frequencies. Among them, small-sized convolutional kernels primarily extract low-frequency components, while large-sized convolutional kernels extract high-frequency components. Not only are the rich image features obtained, but the computational complexity is not increased. Recently, more SR network models have adopted multi-scale learning to improve model performance. In ELAN [79], the authors proposed the grouped multi-scale self-attention (GMSA) module, in which self-attention is computed using windows of different sizes on a non-overlapping set of feature maps, as a way to establish long-range dependencies.

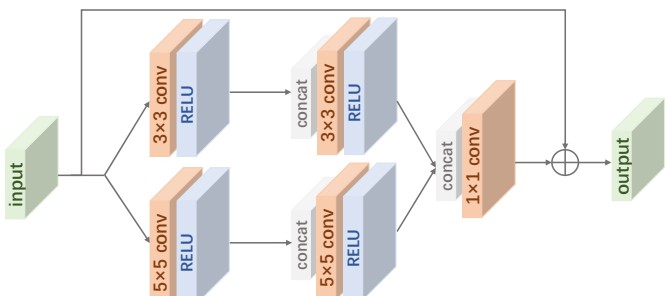

**Figure 6.** The structure of multi-scale residual block (MSRB) [65].

### 3.1.4. Attention Mechanism

The attention mechanism was proposed due to the fact that convolutional neural networks focus more on local information and ignore global features. The attention mechanism is widely used in various computer vision tasks, often inserted into the backbone network as a component, and its main purpose is to allocate computational resources to more important tasks with limited computational power. In short, the attention mechanism helps the network to ignore irrelevant information and focus on important details. Many works have previously been proposed to facilitate the development of attention mechanisms. For example, Hu et al. [80] proposed a novel "squeeze and excite" (SE) block, which adaptively adjusts channel feature responses according to the interdependencies between channels, as shown in Figure 7. With the continuous progress of the attention mechanism and the advancement of previous research work, the attention mechanism has begun to be applied to image super-resolution tasks.

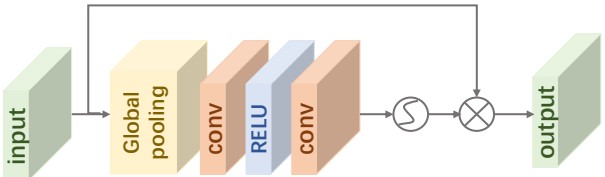

**Figure 7.** The structure of channel attention mechanism [80].

Channel Attention

In RCAN [81], Zhang et al. proposed a residual channel attention block (RCAB) to achieve higher accuracy by learning the correlation between channels to adjust channel features. To make the network pay more attention to the vital spatial features in the residual features, Liu et al. [66] proposed an enhanced spatial attention (ESA) block, which used a $1 \times 1$ convolution to reduce the number of channels to be light enough to be inserted into each residual block. Furthermore, three $3 \times 3$ convolution combinations are used in order to expand the perceptual field. Since channel attention treats each convolutional layer as a separate process and ignores the correlation between different layers, the use of this algorithm will lead to the loss of some intermediate features during the image reconstruction. Therefore, Niu et al. [82] proposed a holistic attention network (HAN) consisting of a layer attention module (LAM) and a channel space attention module (CSAM). The LAM can assign different attention weights to features in different layers by obtaining the dependencies between features of different depths, and then use the CSAM to learn the correlations at different locations in each feature map, so as to capture global features more efficiently. Similarly, the second-order channel attention (SOCA) module in SAN [83] learns the inter-channel feature correlations by using the second-order statistics of the features. The matrix multispectral channel attention (MMCA) module [84] first transformed the image features to the frequency domain by DCT and then learned the channel attention to achieve reconstruction accuracy in the SOTA results.

Non-Local Attention

Due to the limited perceptual field size, most image super-resolution networks are only good at extracting local features in images, ignoring the correlation between long-range features in images. However, this may provide critical information for reconstructing images. Therefore, some studies have been proposed for non-local feature correlation. For example, the purpose of the region non-local RL-NL module in SAN [83] is to divide the input image into specific sizes and perform non-local operations on each region. Liu et al. [85] proposed a non-local recurrent network (NLRN) to introduce non-local operations into recurrent neural networks (RNN) for image recovery tasks to obtain the correlation of deep features at each location with their neighboring features. Regarding non-local attention, a cross-scale non-local (CS-NL) attention module was proposed in CSNLN [86], which computes the similarity between LR feature blocks and HR target feature blocks in an image and improves the performance of the SR model.

Other Attention

In addition to the common attention mechanisms mentioned above, there are also some attention mechanisms designed from a specific perspective. For example, the contextual reasoning attention network [87] generates attention masks based on global contextual information, thus dynamically adjusting the convolutional kernel size to accommodate image feature changes. Zhang et al. [79] argued that the transformer's self-attention computation is too large and certain operations are redundant for super-resolution tasks, so the grouped multi-scale self-attention (GMSA) module was proposed, which computes attention within windows of different sizes while sharing attention to accelerate the computation. Mei et al. [88] introduced sparse representation to non-local self-attention to improve the performance of the attention mechanism and reduce the number of operations.

3.1.5. Feedback Mechanism

The feedback mechanism differs from the input-to-target object mapping by introducing a self-correcting phase to the learning process of the model, i.e., sending the output from the back end to the front end. The feedback mechanism is close to the recursive learning structure, but the difference is that the parameters of the feedback-based model are self-correcting, while the parameters of the recursive learning-based model are shared between modules. In recent years, feedback mechanisms have been gradually applied to

computer vision tasks [89,90]. Feedback mechanisms are also widely used in SR models due to their ability to transfer deep information to the front end of the network to help further the processing of shallow information, which facilitates the reconstruction process from LR images to HR images. Haris et al. [91] proposed a depth inverse projection network for super-resolution, using an alternating upsampling and downsampling stage structure to achieve each stage of error feedback. RBPN [92] was proposed based on DBPN [91] for video super-resolution tasks, which also introduces a feedback mechanism, with the difference that RBPN integrates single-frame input and multi-frame input into one, using an encoder–decoder mechanism to integrate image details. In SFRBN [93], a feedback module (FB) is proposed, where the output of the previous module is fed back to the next module as part of its input, enabling the further refinement of low-level information.

### 3.1.6. Frequency Information-Based Models

In addition to improvements in model volume (increasing width and depth), some scholars have found that many current models for SR have a common problem: feature extraction or processing tends to ignore high-frequency information. The SR task is precisely a process of texture detail reconstruction for LR images, and such a problem can seriously affect the reconstruction results. Therefore, some SR methods that focus on image frequency information have been proposed. Qiu et al. [59] proposed an embedded block residual network in EBRN, which used a recursive approach to the hierarchical processing of features with different frequencies, with low-frequency information processed by a shallow module and high-frequency information processed by a deep module, as a way to achieve better results. Xie et al. [94] proposed a discrete cosine transform-based predictor that partitions the coefficients of the input image in terms of frequency information, thus implementing operations of different complexity for different regions, reducing computational effort and computational complexity. Magid et al. [84] proposed a dynamic high-pass filtering module (HPF) that dynamically adjusts the convolution kernel weights for different spatial locations, thus preserving high-frequency information. A matrix multispectral channel attention (MMCA) module was also proposed, which learns channel attention by transforming features to the frequency domain through DCT. Kong et al. [95] proposed ClassSR consisting of Class-Module and SR-Module to classify and super-resolve the input image based on frequency information. Specifically, the Class-Module first decomposes the image into small sub-images and classifies their complexity, i.e., smooth regions are more accessible to reconstruct than textured regions. Then, these small sub-images are fed to different SR-Module branches for further processing according to different complexity levels.

### 3.1.7. Sparsity-Based Models

In addition to the recovery of high-frequency information, introducing image sparsity into CNN also leads to better performance. In SRN [96], an SR model incorporating sparse coding design was proposed with better performance than SRCNN. Gao et al. [97] presented a hybrid wavelet convolutional network (HWCN) to obtain scattered feature maps by predefined scattering convolution and then the sparse coding of these feature maps, used as the input to the SR model. Wang et al. [98] developed a sparse mask SR (SMSR) network to improve network inference efficiency by teaching sparse masks to cull redundant computations. In SMSR [98], "important" and "unimportant" regions are jointly distinguished by spatial and channel masks, thus skipping unnecessary computations.

### *3.2. Learning Strategies*

Common problems in the training process of SR models based on deep learning include slow convergence and over-fitting. Solutions to these problems are closely related to deep learning strategies, such as selecting the loss function, including regularization, or performing batch normalization. These are critical steps in the training of deep learning models. The purpose of this section is to introduce common learning strategies and optimization algorithms used in deep learning.

### 3.2.1. Loss Function

Loss functions are used to calculate the error between reconstructed images and ground truth. The loss function is a crucial factor in determining the performance of the model and plays a role in guiding the model learning during the training process. A reasonable choice of the loss function can make the model converge faster on the dataset. The smaller the value of the loss function, the better the robustness of the model. In order to better reflect the reconstruction of images, researchers try to use a combination of multiple loss functions (e.g., pixel loss, texture loss, etc.). In this section, we will study several commonly used loss functions.

### Pixel Loss

Pixel loss is the most popular loss function in image super-resolution tasks, which is used to calculate the difference between the reconstructed image and ground truth pixels to make the training process as close to convergence as possible. L1 loss, L2 loss, and Charbonnier loss are among the pixel-level loss functions:

$$L_{L1}(I_{SR}, I_y) = \frac{1}{hwc} \sum_{i,j,k} |I_{SR}^{i,j,k} - I_y^{i,j,k}|, \tag{8}$$

$$L_{L2}(I_{SR}, I_y) = \frac{1}{hwc} \sum_{i,j,k} \left(I_{SR}^{i,j,k} - I_y^{i,j,k}\right)^2, \tag{9}$$

$$L_{Cha}(I_{SR}, I_y) = \frac{1}{hwc} \sum_{i,j,k} \sqrt{\left(I_{SR}^{i,j,k} - I_y^{i,j,k}\right)^2 + \epsilon^2}, \tag{10}$$

where $h$, $w$, and $c$ are the height, width, and number of channels of the image, respectively. $\epsilon$ is a constant (usually set to $10^{-3}$) to ensure stable values. In image super-resolution tasks, many image evaluation metrics involve inter-pixel differences, such as PSNR, so pixel loss has been a popular loss function in super-resolution. However, pixel loss does not consider the perceptual quality and texture of the reconstructed image, which can lead to a lack of reconstructed images with lost high-frequency details; therefore, high-quality reconstructed images cannot be obtained.

### Perceptual Loss

Perceptual loss is commonly used in GAN networks. In order to obtain reconstructed images with rich high-frequency features, researchers proposed perceptual loss in place of the L2 loss used previously to calculate inter-pixel differences. Specifically, perceptual loss is often used to compare two images that look similar but are different, because perceptual loss compares the perceptual quality and semantic differences between the reconstructed image and ground truth:

$$L_{\text{perceptual}} = \frac{1}{h_j w_j c_j} \left\| \varnothing_l(I_{SR}) - \varnothing_l(I_y) \right\|_2^2, \tag{11}$$

where $h_l$, $w_l$, and $c_l$ denote the height, width, and number of channels of the *l*-th layer features, respectively. $\varnothing$ denotes the pre-trained network, and $\varnothing_{(l)}(I)$ denotes the high-level features extracted from a certain *l*-th layer of the network.

### Content Loss

Content loss was applied early in the field of style migration, and is similar to perceptual loss, using the semantic difference between the generated and content images compared with the trained classification network, i.e., L2 distance:

$$L_{\text{Content}}(I_{SR}, I_y, \varnothing, l) = \frac{1}{h_l w_l c_l} \sum_{i,j,k} \left( \varnothing_{(l)}^{i,j,k}(I_{SR}) - \varnothing_{(l)}^{i,j,k}(I_y) \right), \tag{12}$$

where $h_l$, $w_l$, and $c_l$ denote the height, width, and number of channels of the *l*-th layer features, respectively. $\varnothing$ denotes the pre-trained classification network, and $\varnothing_{(l)}(I)$ denotes the high-level features extracted from the *l*-th layer of the network.

Texture Loss

By obtaining the spatial correlation between the feature maps in the pre-trained network, texture loss is a modification of perceptual loss as introduced by Gatys et al. to the field of style migration. Since the reconstructed images possess the same style as ground truth, texture loss can also be applied in the field of super-resolution [14,99–101]. Texture loss is mainly achieved by computing the Gram matrix:

$$G^{ij}_{(l)}(I) = \text{vec}\left(\varnothing^i_{(l)}(I)\right) \cdot \text{vec}\left(\varnothing^j_{(l)}(I)\right), \tag{13}$$

where $G^{ij}_{(l)}(I)$ is the inner product between vectorized feature maps *i* and *j* at the *l*-th layer, which captures the tendency of features to appear simultaneously in different parts of the image. vec() denotes the vectorization operation and $\varnothing^i_{(l)}(I)$ denotes the *i*-th channel of the feature map on the *l*-th layer of image *I*. Then, the texture loss is defined as follows:

$$L_{\text{texture}}\left(I_{SR}, I_y, \varnothing, l\right) = \frac{1}{c_l^2} \sqrt{\sum_{i,j}\left(G^{ij}_{(l)}(I_{SR}) - G^{ij}_{(l)}(I_y)\right)^2}. \tag{14}$$

Adversarial Loss

Recent research has demonstrated that generative adversarial networks (GANs) perform well on image super-resolution tasks. GANs are gradually being applied to computer vision tasks. A generative adversarial network (GAN) consists of two core parts: generator and discriminator. It is the generator's responsibility to create data that do not exist, while the discriminator is responsible for determining whether the generated data are accurate or false. After iterative training, the ultimate goal of the generator is to generate data that look naturally real and are as close to the original data as possible, so that the discriminator cannot determine the authenticity. The task of the discriminative model is to identify the fake data as accurately as possible, and the application of GAN in the field of super-resolution takes the form of SRGAN [102]. The design of the SRGAN loss function is based on the cross-entropy in pixel loss, which is defined as follows:

$$L_{\text{Adversarial}}\left(I_x, G, D\right) = \sum_{n=1}^{N} -\log D(G(I_x)), \tag{15}$$

where $G(I_x)$ is the reconstructed SR image, and *G* and *D* represent the generator and discriminator, respectively. Some MOS tests have shown that SR models trained by a combination of content loss and adversarial loss perform better in terms of the perceptual quality of images than SR models that undergo only pixel loss. Still, with reduced PSNR, research continues on how to integrate GAN into SR models and stabilize the trained GAN.

3.2.2. Regularization

The SR model training process is prone to the over-fitting phenomenon; that is, the model overlearns the training dataset and has poor generalization ability, resulting in a high evaluation index of the reconstructed image output on the training set and poor performance on the test set. The reasons for over-fitting include (1) the small size of the training dataset, and (2) the high complexity of the model and numerous parameters. Therefore, the most direct way to avoid overfitting is to increase the size of the training dataset so that the training set samples are as close as possible to the ground-truth data distribution. However, this approach does not guarantee the effect and is time-consuming

and laborious. In deep learning, a common learning strategy used to prevent overfitting is regularization.

The essence of regularization is to preserve the original features, make the input dataset smaller than the number of model parameters, and avoid training, in order to obtain parameters that improve the generalization ability of the model and prevent overfitting. The common regularization methods [103,104] in deep learning include L1\L2 regularization, dropout [105–107], early stopping, and data augmentation.

L1\L2 Regularization

L1\L2 regularization is the most commonly used regularization method. It essentially adds regular terms of L1\L2 parameterization to the loss function to reduce the number of parameters, acting as a parameter penalty to reduce the complexity of the model and limit its learning ability.

L1 regularization is defined as follows:

$$cost = Loss + \gamma \sum \|w\|, \tag{16}$$

L2 regularization is defined as follows:

$$cost = Loss + \gamma \sum \|w\|^2, \tag{17}$$

where $\gamma$ is the hyperparameter that controls the proportion of the control loss term and the regularization term, and $w$ is the model weight.

Since many parameter vectors in the L1 regularization term are sparse vectors, resulting in many parameters being zero after model regularization, L1 regularization is used when compressing the model in deep learning, while L2 regularization is commonly used in other cases.

Dropout

Hinton et al. [108] proposed that when the dataset is small, and the neural network model is large and complex, over-fitting tends to occur during training. To prevent overfitting, some of the feature detectors can be stopped in each training batch so that the model does not rely too much on certain local features, thus improving the generalization ability and performance of the model. Compared with other regularization methods, dropout [109] is simpler to implement, has essentially no restrictions on the model structure, and has good performance on feedforward neural networks, probabilistic models, and recurrent neural networks, with a wide range of applicability. There are two typical dropout implementations: vanilla dropout and inverted dropout.

The process of vanilla dropout includes the model being trained by randomly dropping some neurons with a certain probability $p$. Then, forward propagation is performed, the loss is calculated, and backward propagation and gradient update are performed. Finally, the step of randomly dropping neurons is repeated. However, the selection of neurons is random for each dropout, and vanilla dropout requires scaling (i.e., multiplying by $(1 - p)$) the trained parameters at test time, which leads to different results for each test with the same input, making the model performance unstable and the operation of balancing expectations too cumbersome. Therefore, vanilla dropout is not widely used.

Inverted dropout is an improved version of vanilla dropout. It is based on the principle of dropping a portion of neurons with random probability $p$ during the model training process. Unlike vanilla dropout, it does not process the parameters during the test stage. Inverted dropout scales the output values by a factor of $\frac{1}{1-p}$ during forward propagation, balancing the expectation values and keeping the training and testing processes consistent.

Early Stopping

As the number of training iterations increases, the training error of the model gradually decreases but the error in the test set increases again. The strategy of stopping the algorithm when the error on the test set does not improve further within a pre-specified number

of cycles, at which point the parameters of the model are stored, and the parameter that minimizes the error on the test set is returned, is called early termination [110,111]. The early termination method hardly changes the model training parameters and optimization objectives and does not disrupt the learning process of the model. Due to its outstanding effectiveness and simplicity, the early termination method is the more commonly used regularization method.

Data Augmentation

Training with a larger number of datasets is the most direct way to improve model generalization and prevent over-fitting. Furthermore, data augmentation [112,113] is an important method to meet the demand of deep learning models for large amounts of data. In general, the size of a dataset is fixed, and data augmentation increases the amount of data by manually generating new data. For images, a single image can be flipped, rotated, cropped, or even Gaussian blurred to generate other forms of images.

### 3.2.3. Batch Normalization

To address the problem of the data distribution within a deep network changing during training, Sergey et al. [114] proposed batch normalization (BN) to avoid covariance shifts within parameters during training. Batch normalization is introduced into the deep learning network framework as a layer, commonly used after the convolution layer, to readjust the data distribution. The BN layer divides the input data into batches, a batch being the number of samples optimized each time, in order to calculate the mean and variance of the groups, and then normalizes them, since each group determines the gradient and reduces randomness when descending. Finally, scaling and offset operations are performed on the data to achieve a constant transformation, and the data recover their original distribution.

Batch normalization can prevent over-fitting from appearing to a certain extent, which is similar to the effect of dropout and improves the generalization ability of the model. Meanwhile, because batch normalization normalizes the mean and variance of parameters in each layer, it solves the problem of gradient disappearance. It supports the use of a larger learning rate, which increases the magnitude of gradient dropout and increases the training speed.

### *3.3. Other Improvement Methods*

In addition to the network design strategies mentioned in Section 3.1, this section will add other design approaches that have further research value.

### 3.3.1. Knowledge-Distillation-Based Models

Hinton et al. [115] first introduced the concept of knowledge distillation, a model compression algorithm based on a "teacher–student network", where the critical problem is how to transfer the knowledge transformed from a large model (teacher model) to a small model (student model). Lee et al.[116] proposed a distillation structure for SR, which was the first time that knowledge distillation was introduced into the super-resolution domain. Knowledge distillation has been widely used in various computer vision tasks, and its advantages of saving computational and storage costs have been shown. In [116], features from the decoder of the teacher network are transferred to the student network in the form of feature distillation, which enables the student network to learn richer detailed information. Zhang et al. [117] proposed a network distillation method DAFL applicable to cell phones and smart cameras in the absence of raw data, using a GAN network to simulate the raw training data in the teacher network, and using a progressive distillation strategy to distill more information from the teacher network and better train the student network.

### 3.3.2. Adder-Operation-Based Models

Nowadays, the convolution operation is a common step in deep learning, and the primary purpose of convolution is to calculate the correlation between the input features

and the filter, which will result in a large number of floating-point-valued multiplication operations. To reduce the computational cost, Chen et al. [118] proposed to use additive operations instead of multiplication operations in convolutional neural networks, i.e., L1 distance is used instead of convolution to calculate correlation, while L1-norm is used to calculate variance, and an adaptive learning rate scale change strategy is developed to speed up model convergence. Due to the superior results produced by AdderNet, Chen et al. [119] applied the additive operation to the image super-resolution task. In AdderSR [119], the relationship between adder operation and constant mapping is analyzed, and a shortcut is inserted to stabilize the performance. In addition, a learnable power activation is proposed to emphasize high-frequency information.

### 3.3.3. Transformer-Based Models

In recent years, the excellence of transformer in the field of natural language processing has driven its application in computer vision tasks. Many transformer-based image processing methods have been proposed one after another, e.g., image classification [120,121], image segmentation [122,123], etc. The advantage of transformer is that self-attention can model long-term dependencies in images [124] and obtain high-frequency information, which helps to recover the texture details of images. Yang et al. [101] proposed a texture transformer network for image super-resolution, where the texture transformer of the method extracts texture information based on the reference image and transfers it to the high-resolution image while fusing different levels of features in a cross-scale manner, obtaining better results compared with the latest methods. Chen et al. [125] proposed a hybrid attention transformer that improves the ability to explore pixel information by introducing channel attention into the transformer while proposing an overlapping cross-attention module (OCAB) to better fuse features from different windows. Lu et al. [126] proposed an efficient and lightweight super-resolution CNN combined with transformer (ESRT), where, on the one hand, the feature map is dynamically resized by the CNN part to extract deep features. On the other hand, the long-term dependencies between similar patches in an image are captured by the efficient transformer (ET) and efficient multi-headed attention (EMHA) mechanisms to save computational resources while improving model performance. The transformer combined with CNN for SwinIR [127] can be used for super-resolution reconstruction to learn the long-term dependencies of images using a shifted window mechanism. Cai et al. [128] proposed a hierarchical patch transformer, which is a hierarchical partitioning of the patches of an image for different regions, for example, using smaller patches for texture-rich regions of the image, to gradually reconstruct high-resolution images.

Transformer-based SR methods are quickly evolving and are being widely adopted due to their superior results, but their large number of parameters and the required amount of computational effort are still problems to be solved.

### 3.3.4. Reference-Based Models

The proposed reference-based SR method alleviates the inherent pathological problem of SR, i.e., an LR image can be obtained by degrading multiple HR images. RefSR used external images from various sources (e.g., cameras, video frames, and network images) as a reference to improve data diversity while conveying reference features and providing complementary information for the reconstruction of LR images. Zhang et al. [99] proposed that the previous RefSR suffers from the problem that the reference image is required to have similar content to the LR image, otherwise it will affect the reconstruction results. To solve the above problems, SRNTT [99] borrowed the idea of neural texture migration for semantically related features after matching the features of the LR image and reference image. In TTSR [101], Yang et al. proposed a texture transformer based on the reference image to extract the texture information of the reference image and transfer it to the high-resolution image.

## 4. Analyses and Comparisons of Various Models

### 4.1. Details of the Representative Models

To describe the performance of the SR models mentioned in Section 3 more intuitively, 16 of these representative models are listed in Table 2, including their PSNR and SSIM metrics on Set5 [35], Set14 [36], BSD100 [32], Urban100 [37], and Manga109 [46] datasets, training datasets, and the number of parameters (i.e., model size).

**Table 2.** PSNR/SSIM comparison on Set5 [35], Set14 [36], BSD100 [32], Urban100 [37], and Manga109 [46]. In addition, the number of training datasets and the parameters of the model are provided.

| Models | Scale | Set5 PSNR/SSIM | Set14 PSNR/SSIM | BSD100 PSNR/SSIM | Urban100 PSNR/SSIM | Manga109 PSNR/SSIM | Train Data | Param. |
|---|---|---|---|---|---|---|---|---|
| SRCNN [18] | ×2 | 36.66/0.9542 | 32.45/0.9067 | 31.36 0.8879 | 29.50/0.8946 | 35.60/0.9663 | T91 + ImageNet | 57 K |
| VDSR [21] | ×2 | 37.53/0.9587 | 33.03/0.9124 | 31.90/0.8960 | 30.76/0.9140 | -/- | BSD + T91 | 665 K |
| DRCN [19] | ×2 | 37.63/0.9588 | 33.04/0.9118 | 31.85/0.8942 | 30.75/0.9133 | -/- | T91 | 1.8 M |
| DRRN [57] | ×2 | 37.74/0.9591 | 33.23/0.9136 | 32.05/0.8973 | 31.23/0.9188 | -/- | BSD + T91 | 297 K |
| CARN [58] | ×2 | 37.76/0.9590 | 33.52/0.9166 | 32.09/0.8978 | 31.92/0.9256 | -/- | BSD + T91 + DIV2K | 1.6 M |
| EDSR [68] | ×2 | 38.11/0.9601 | 33.92/0.9195 | 32.32/0.9013 | 32.93/0.9351 | -/- | DIV2K | 43 M |
| ELAN [79] | ×2 | 38.17/0.9611 | 33.94/0.9207 | 32.30/0.9012 | 32.76/0.9340 | 39.11/0.9782 | DIV2K | 8.3 M |
| MSRN [65] | ×2 | 38.08/0.9605 | 33.74/0.9170 | 32.23/0.9013 | 32.22/0.9326 | 38.82/0.9868 | DIV2K | 6.5 M |
| RCAN [81] | ×2 | 38.27/0.9614 | 34.12/0.9216 | 32.41/0.9027 | 33.34/0.9384 | 39.44/0.9786 | DIV2K | 16 M |
| HAN [82] | ×2 | 38.27/0.9614 | 34.16/0.9217 | 32.41/0.9027 | 33.35/0.9385 | 39.46/0.9785 | DIV2K | 16.1 M |
| RDN [64] | ×2 | 38.30/0.9616 | 34.10/0.9218 | 32.40/0.9022 | 33.09/0.9368 | 39.38/0.9784 | DIV2K | 22.6 M |
| NLSN [88] | ×2 | 38.34/0.9618 | 34.08/0.9231 | 32.43/0.9027 | 33.42/0.9394 | 39.59/0.9789 | DIV2K | 16.1 M |
| RFANet [66] | ×2 | 38.26/0.9615 | 34.16/0.9220 | 32.41/0.9026 | 33.33/0.9389 | 39.44/0.9783 | DIV2K | 11 M |
| SAN [83] | ×2 | 38.31/0.9620 | 34.07/0.9213 | 32.42/0.9028 | 33.10/0.9370 | 39.32/0.9792 | DIV2K | 15.7 M |
| SMSR [98] | ×2 | 38.00/0.9601 | 33.64/0.9179 | 32.17/0.8990 | 32.19/0.9284 | 38.76/0.9771 | DIV2K | 1 M |
| ESRT [126] | ×2 | -/- | -/- | -/- | -/- | -/- | DIV2K | 751 K |
| TDPN [14] | ×2 | 38.31/0.9621 | 34.16/0.9225 | 32.52/0.9045 | 33.36/0.9386 | 39.57/0.9795 | DIV2K | 12.8 M |
| SwinIR [127] | ×2 | 38.42/0.9623 | 34.46/0.9250 | 32.53/0.9041 | 33.81/0.9427 | 39.92/0.9797 | DIV2K + Flickr2K | 12 M |
| SRCNN [18] | ×3 | 32.75/0.9090 | 29.30/0.8215 | 28.41/0.7863 | 26.24/0.7989 | 30.48/0.9117 | T91 + ImageNet | 57 K |
| VDSR [21] | ×3 | 33.66/0.9213 | 29.77/0.8314 | 28.82/0.7976 | 27.14/0.8279 | -/- | BSD + T91 | 665 K |
| DRCN [19] | ×3 | 33.82/0.9226 | 29.76/0.8311 | 28.80/0.7963 | 27.15/0.8276 | -/- | T91 | 1.8 M |
| DRRN [57] | ×3 | 34.03/0.9244 | 29.96/0.8349 | 28.95/0.8004 | 27.53/0.8378 | -/- | BSD + T91 | 297 K |
| CARN [58] | ×3 | 34.29/0.9255 | 30.29/0.8407 | 29.06/0.8034 | 28.06/0.8493 | -/- | BSD + T91 + DIV2K | 1.6 M |
| EDSR [68] | ×3 | 34.65/0.9282 | 30.52/0.8462 | 27.71/0.7420 | 29.25/0.8093 | -/- | DIV2K | 43 M |
| ELAN [79] | ×3 | 34.61/0.9288 | 30.55/0.8463 | 29.21/0.8081 | 28.69/0.8624 | 34.00/0.9478 | DIV2K | 8.3 M |
| MSRN [65] | ×3 | 34.38/0.9262 | 30.34/0.8395 | 29.08/0.8041 | 28.08/0.8554 | 33.44/0.9427 | DIV2K | 6.5 M |
| RCAN [81] | ×3 | 34.74/0.9299 | 30.65/0.8482 | 29.32/0.8111 | 29.09/0.8702 | 34.44/0.9499 | DIV2K | 16 M |
| HAN [82] | ×3 | 34.75/0.9299 | 30.67/0.8483 | 29.32/0.8110 | 29.10/0.8705 | 34.48/0.9500 | DIV2K | 16.1 M |
| RDN [64] | ×3 | 34.78/0.9300 | 30.67/0.8482 | 29.33/0.8105 | 29.00/0.8683 | 34.43/0.9498 | DIV2K | 22.6 M |
| NLSN [88] | ×3 | 34.85/0.9306 | 30.70/0.8485 | 29.34/0.8117 | 29.25/0.8726 | 34.57 0.9508 | DIV2K | 16.1 M |
| RFANet [66] | ×3 | 34.79/0.9300 | 30.67/0.8487 | 29.34/0.8115 | 29.15/0.8720 | 34.59/0.9506 | DIV2K | 11 M |
| SAN [83] | ×3 | 34.75/0.9300 | 30.59/0.8476 | 30.59/0.8476 | 28.93/0.8671 | 34.30/0.9494 | DIV2K | 15.7 M |
| SMSR [98] | ×3 | 34.40/0.9270 | 30.33/0.8412 | 29.10/0.8050 | 28.25/0.8536 | 33.68/0.9445 | DIV2K | 1 M |
| ESRT [126] | ×3 | 34.42/0.9268 | 30.43/0.8433 | 29.15/0.8063 | 28.46/0.8574 | 33.95/0.9455 | DIV2K | 751 K |
| TDPN [14] | ×3 | 34.86/0.9312 | 30.79/0.8501 | 29.45/0.8126 | 29.26/0.8724 | 34.48/0.9508 | DIV2K+Flickr2K | 12.8 M |
| SwinIR [127] | ×3 | 34.97/0.9318 | 30.93/0.8534 | 29.46/0.8145 | 29.75/0.8826 | 35.12/0.9537 | DIV2K + Flickr2K | 12 M |
| SRCNN [18] | ×4 | 30.48/0.8628 | 27.50/0.7513 | 26.90/0.7101 | 24.52/0.7221 | 27.58/0.8555 | T91 + ImageNet | 57 K |
| VDSR [21] | ×4 | 31.35/0.8838 | 28.01/0.7674 | 27.29/0.7260 | 25.18/0.7524 | -/- | BSD + T91 | 665 K |
| DRCN [19] | ×4 | 31.53/0.8854 | 28.02/0.7670 | 27.23/0.7233 | 25.14/0.7510 | -/- | T91 | 1.8 M |
| DRRN [57] | ×4 | 31.68/0.8888 | 28.21/0.7720 | 27.38/0.7284 | 25.44/0.7638 | -/- | BSD + T91 | 297 K |
| CARN [58] | ×4 | 32.13/0.8937 | 28.60/0.7806 | 27.58/0.7349 | 26.07/0.7837 | -/- | BSD + T91 + DIV2K | 1.6 M |
| EDSR [68] | ×4 | 32.46/0.8968 | 28.80/0.7876 | 27.71/0.7420 | 26.6 /0.8033 | -/- | DIV2K | 43M |
| ELAN [79] | ×4 | 32.43/0.8975 | 28.78/0.7858 | 27.69/0.7406 | 26.54/0.7982 | 30.92/0.9150 | DIV2K | 8.3 M |
| MSRN [65] | ×4 | 32.07/0.8903 | 28.60/0.7751 | 27.52/0.7273 | 26.04/0.7896 | 30.17/0.9034 | DIV2K | 6.5 M |
| RCAN [81] | ×4 | 32.63/0.9002 | 28.87/0.7889 | 27.77/0.7436 | 26.82/0.8087 | 31.22/0.9173 | DIV2K | 16 M |
| HAN [82] | ×4 | 32.64/0.9002 | 28.90/0.7890 | 27.80/0.7442 | 26.85/0.8094 | 31.42/0.9177 | DIV2K | 16.1 M |
| RDN [64] | ×4 | 32.61/0.9003 | 28.92/0.7893 | 27.80/0.7434 | 26.82/0.8069 | 31.39/0.9184 | DIV2K | 22.6 M |
| NLSN [88] | ×4 | 32.59 0.9000 | 28.87 0.7891 | 27.78 0.7444 | 26.96 0.8109 | 31.27 0.9184 | DIV2K | 16.1 M |
| RFANet [66] | ×4 | 32.66/0.9004 | 28.88/0.7894 | 27.79/0.7442 | 26.92/0.8112 | 31.41/0.9187 | DIV2K | 11 M |
| SAN [83] | ×4 | 32.64/0.9003 | 28.92/0.7888 | 27.78/0.7436 | 26.79/0.8068 | 31.18/0.9169 | DIV2K | 15.7 M |
| SMSR [98] | ×4 | 32.12/0.8932 | 28.55/0.7808 | 27.55/0.7351 | 26.11/0.7868 | 30.54/0.9085 | DIV2K | 1 M |
| ESRT [126] | ×4 | 32.19/0.8947 | 28.69/0.7833 | 27.69/0.7379 | 26.39/0.7962 | 30.75/0.9100 | DIV2K | 751 K |
| TDPN [14] | ×4 | 32.69/0.9005 | 29.01/0.7943 | 27.93/0.7460 | 27.24/0.8171 | 31.58/0.9218 | DIV2K | 12.8 M |
| SwinIR [127] | ×4 | 32.92/0.9044 | 29.09/0.7950 | 27.92/0.7489 | 27.45/0.8254 | 32.03/0.9260 | DIV2K + Flickr2K | 12 M |

By comparing them, the following conclusions can be drawn: (1) To better visualize the performance differences between these models, we selected the number of parameters and the PSNR metrics of these models on the Set5 dataset and plotted a line graph, as shown in Figure 8. Usually, the larger the number of parameters, the better the reconstruction results, but this does not show that increasing the model size will improve the model performance, which is inaccurate. (2) Without considering the model size, the image super-resolution used for the transformer models tends to perform well. (3) Lightweight (that is, the number of parameters is less than 1000 K) and efficient models are in the minority in the field of image super-resolution, but in the future will become the mainstream direction of research.

Additionally, we list some classical methods, datasets, and evaluation metrics of remote sensing image super-resolution models in Table 3, sorted by year of publication. In analyzing the data, we can observe that, on the one hand, research methods in RSISR are gradually diversifying and have improved in terms of their performance in recent years. On the other hand, less attention is being paid to research on large-scale remote sensing super-resolution methods, which represents an area in which research will be challenging.

**Table 3.** PSNR/SSIM of some representative methods for remote sensing image super-resolution.

| Models | Method | Scale | Dataset | PSNR/SSIM |
|---|---|---|---|---|
| LGCnet [22] | combination of local and global Information | ×2 ×3 ×4 | UC Merced | 33.48/0.9235 29.28/0.8238 27.02/0.7333 |
| RS-RCAN [129] | residual channel attention | ×2 ×3 ×4 | UC Merced | 34.37/0.9296 30.26/0.8507 27.88/0.7707 |
| WTCRR [130] | wavelet transform, recursive learning and residual learning | ×2 ×3 ×4 | NWPU-RESISC45 | 35.47/0.9586 31.80/0.9051 29.68/0.8497 |
| CSAE [131] | sparse representation and coupled sparse autoencoder | ×2 ×3 | NWPU-RESISC45 | 29.070/0.9343 25.850/0.8155 |
| DRGAN [132] | a dense residual generative adversarial | ×2 ×3 ×4 | NWPU-RESISC45 | 35.56/0.9631 31.92/0.9102 29.76/0.8544 |
| MPSR [133] | enhanced residual block (ERB) and residual channel attention group(RCAG) | ×2 ×3 ×4 | UC Merced | 39.78/0.9709 33.93/0.9199 30.34/0.8584 |
| RDBPN [134] | residual dense backprojection network | ×4 ×8 | UC Merced | 25.48/0.8027 21.63/0.5863 |
| EBPN [135] | enhanced back-projection network(EBPN) | ×2 ×4 ×8 | UC Merced | 39.84/0.9711 30.31/0.8588 24.13/0.6571 |
| CARS [136] | channel attention | ×4 | Pleiades1A | 30.8971/0.9489 |
| FeNet [137] | a lightweight feature enhancement network) | ×2 ×3 ×4 | UC Merced | 34.22/0.9337 29.80/0.8481 27.45/0.7672 |

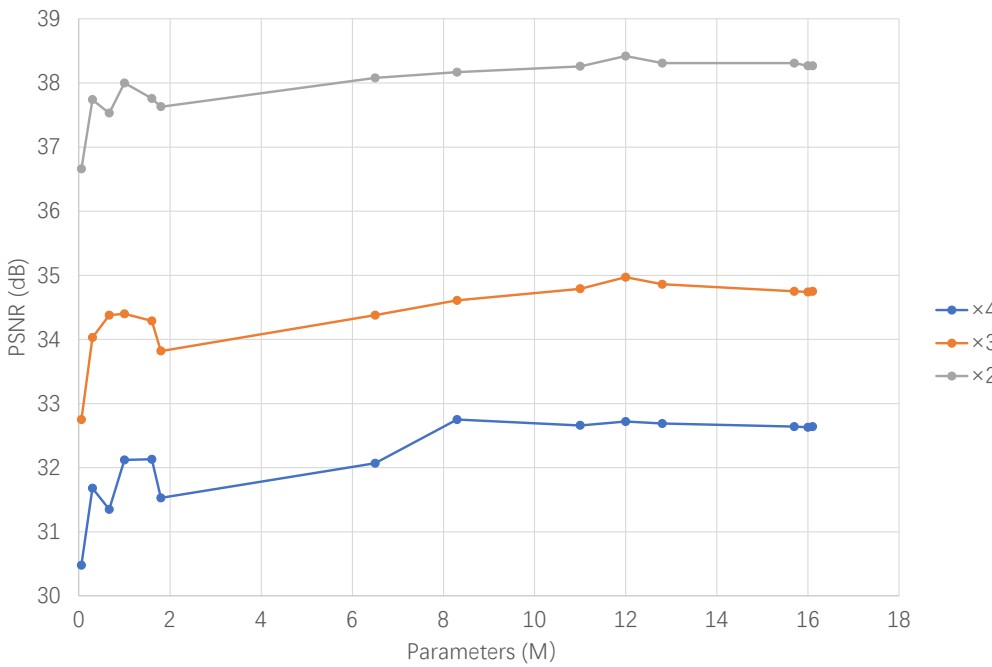

**Figure 8.** Variation of PSNR with the number of parameters.

### 4.2. Results and Discussion

To visualize the results of our experiments on remote sensing image datasets, we select classical SR models and present the visualization results (Figure 9) to visually and comprehensively illustrate their application on remote sensing images. In particular, we retrain these models and test them based on the WHU-RS19 [40] and RSC11 [44] datasets.

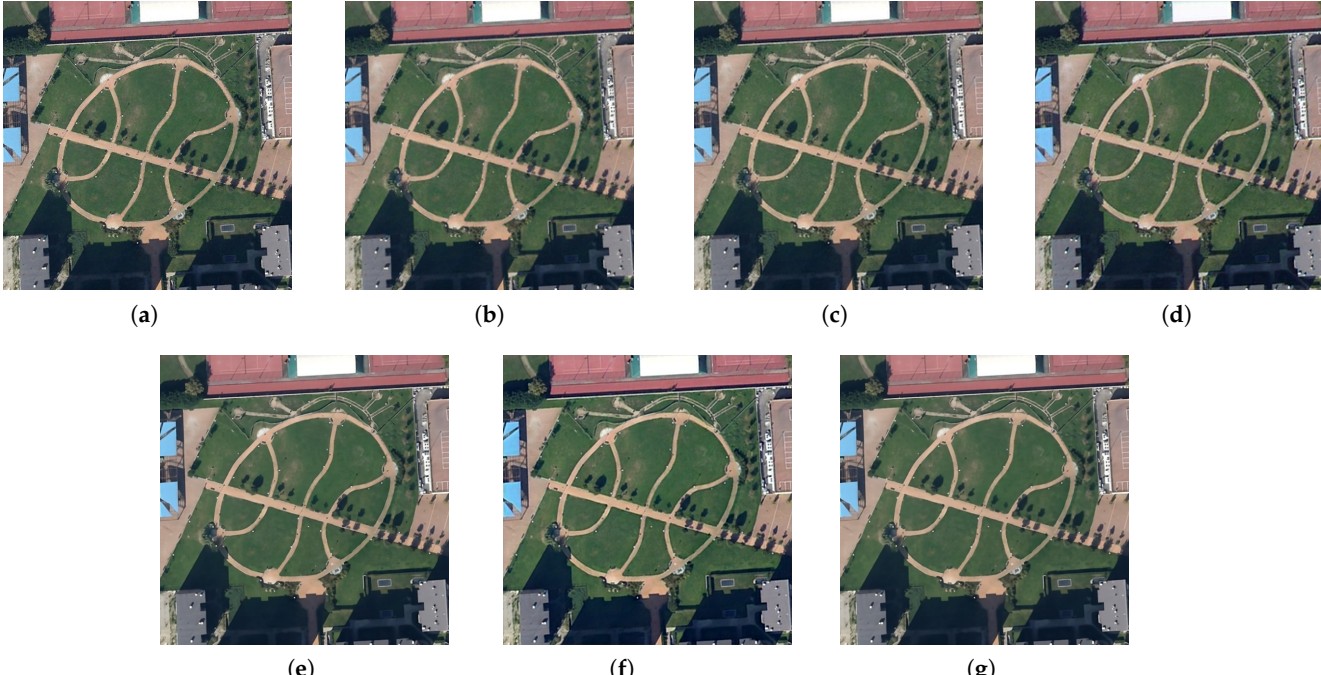

**Figure 9.** Comparison of visual results of different SR methods with ×2 super-resolution on the WHU-RS19 [40] dataset (square scene). (**a**) HR. (**b**) Bicubic. (**c**) EDSR [68]. (**d**) RCAN [81]. (**e**) RDN [64]. (**f**) SAN [83]. (**g**) NLSN [88].

Figure 9 illustrates the comparison of different SR methods for the super-resolution reconstruction of the WHU-RS19 [40] dataset from remote sensing images. When compared with the HR images, the results obtained by bicubic interpolation and EDSR [68] all exhibit a loss of detail and a smoothing effect. NLSN [88] appears to retain high-frequency information better, with the texture details of the reconstructed images being close to those of HR images, and the contours of the structures in the images being more clearly defined.

Figure 10 shows the results of the SR method for ×2 super-resolution reconstruction on a parking lot image in the WHU-RS19 [40] dataset. There are a variety of car colors present in the scene. Color shifts are observed using both the bicubic interpolation and RCAN [81] methods. RDN [64] with dense residual blocks provides accurate color results. The results of all other reconstruction methods are blurry.

The results of the SR method for ×2 super-resolution reconstruction on the WHU-RS19 [40] dataset from forests are given in Figure 11. Except for SAN [83] and RCAN [81], all other methods show high color similarity to the HR image. The results of several attention-based methods are also acceptable in terms of texture features, and the edge details of the forest are relatively well-defined.

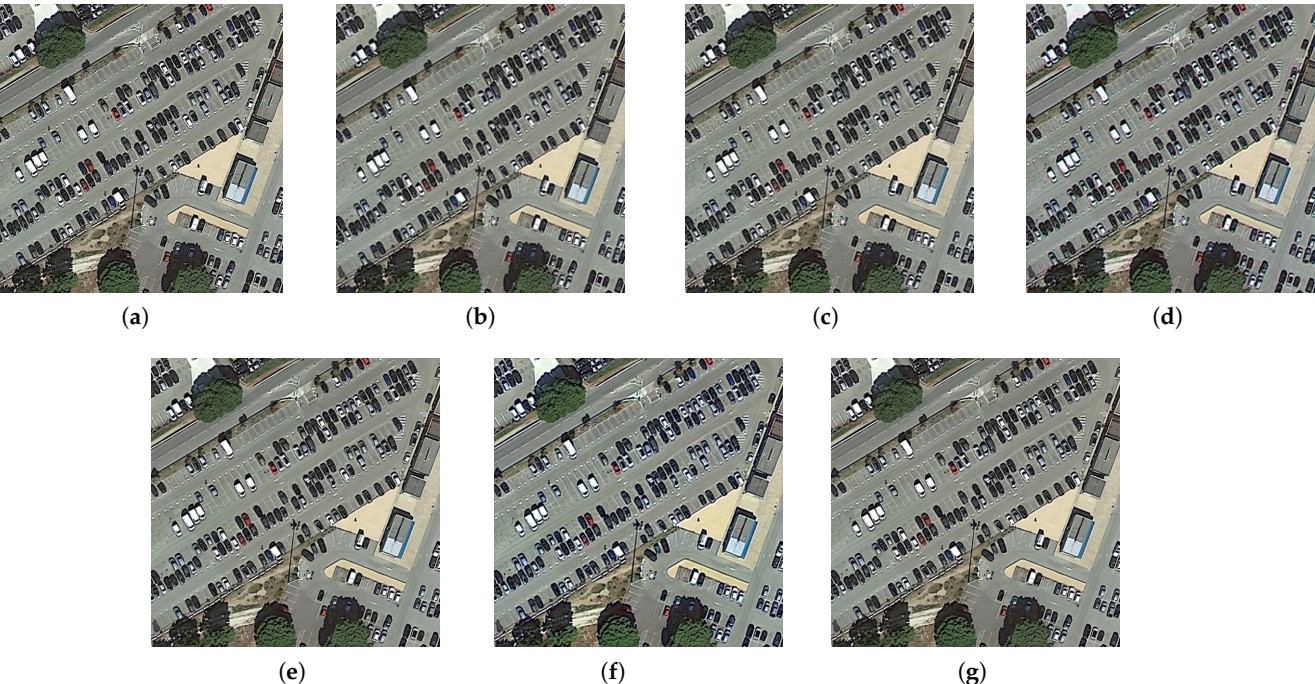

**Figure 10.** Comparison of visual results of different SR methods with ×2 super-resolution on the WHU-RS19 [40] dataset (parking lot scene). (**a**) HR. (**b**) Bicubic.(**c**) EDSR [68]. (**d**) RCAN [81]. (**e**) RDN [64]. (**f**) SAN [83]. (**g**) NLSN [88].

Figure 12 shows the results of the SR method for ×2 super-resolution reconstruction on the port images in the RSC11 [44] dataset. SAN [83] and RDN [64] methods provide better visual results both in terms of spatial and spectral characteristics. It is easier to identify objects such as boats in the scene based on the reconstruction results. EDSR [68] and bicubic interpolation results are blurrier around the edges.

Figure 13 shows the effect of the SR method on the ×2 super-resolution reconstruction of the residential area images in the RSC [44] dataset. In the reconstruction results of the CNN-based SR method, some exterior contours of the buildings can be observed, and useful geometric features are retained. The result of the bicubic interpolation process is blurrier and lacks some spatial detail features.

Figure 14 shows the results of the SR method for ×2 super-resolution reconstruction on sparse forest images in the RSC11 [44] dataset. The result generated by NLSN [88] is closer

to the color characteristics of HR and better preserves the color of the plain land. RDN [64] retains more texture features and can observe detailed information such as branches and trunks of trees.

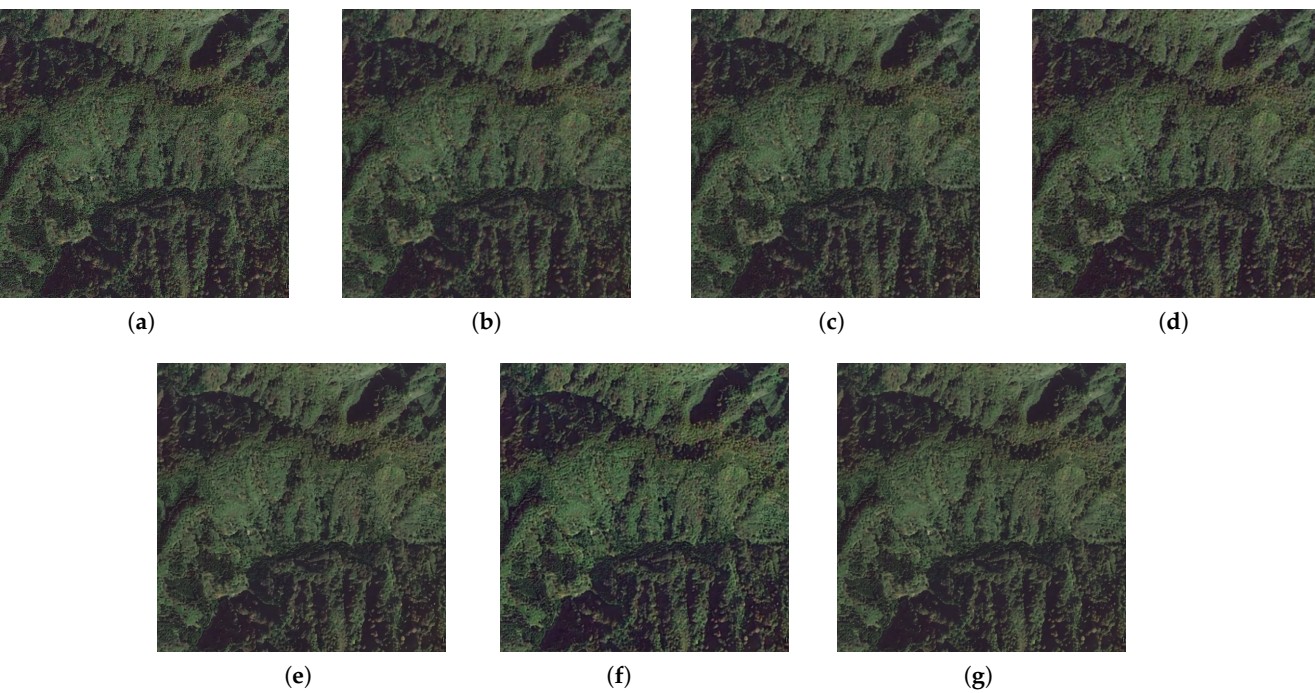

**Figure 11.** Comparison of visual results of different SR methods with ×2 super-resolution on the WHU-RS19 [40] dataset (forest scene). (**a**) HR. (**b**) Bicubic. (**c**) EDSR [68]. (**d**) RCAN [81]. (**e**) RDN [64]. (**f**) SAN [83]. (**g**) NLSN [88].

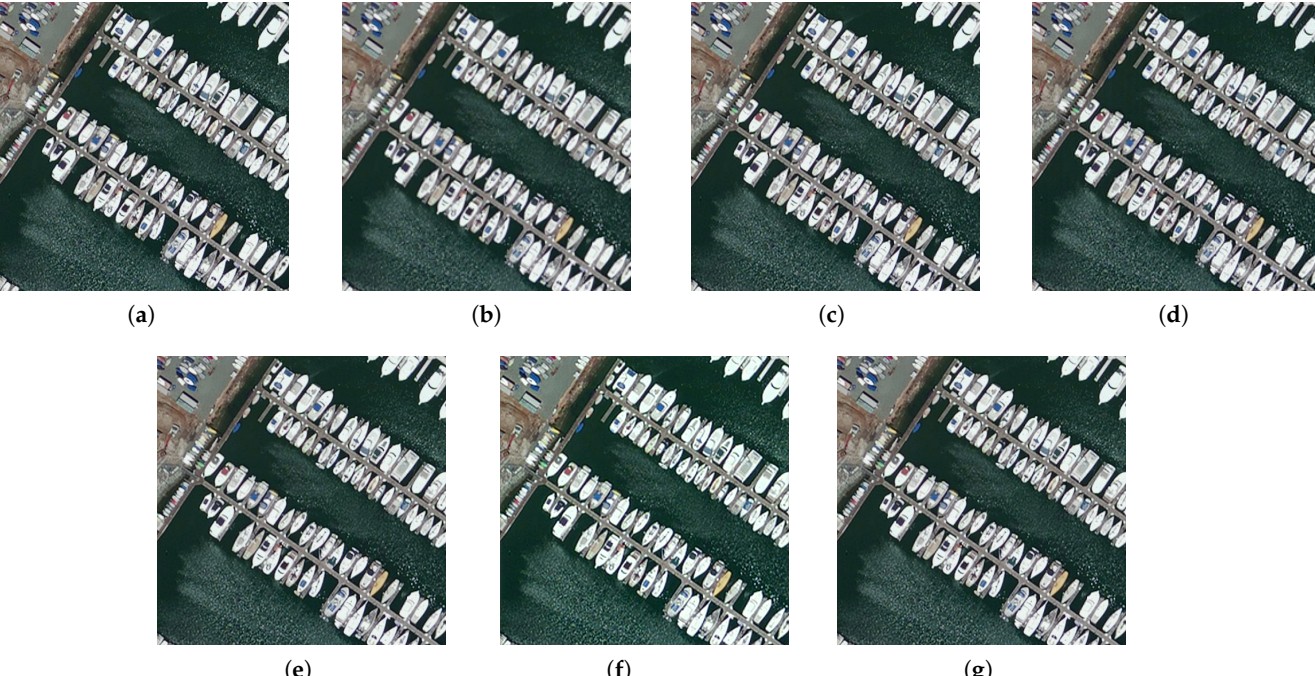

**Figure 12.** Comparison of visual results of different SR methods with ×2 super-resolution on the RSC11 [44] dataset (port scene). (**a**) HR. (**b**) Bicubic. (**c**) EDSR [68]. (**d**) RCAN [81]. (**e**) RDN [64]. (**f**) SAN [83]. (**g**) NLSN [88].

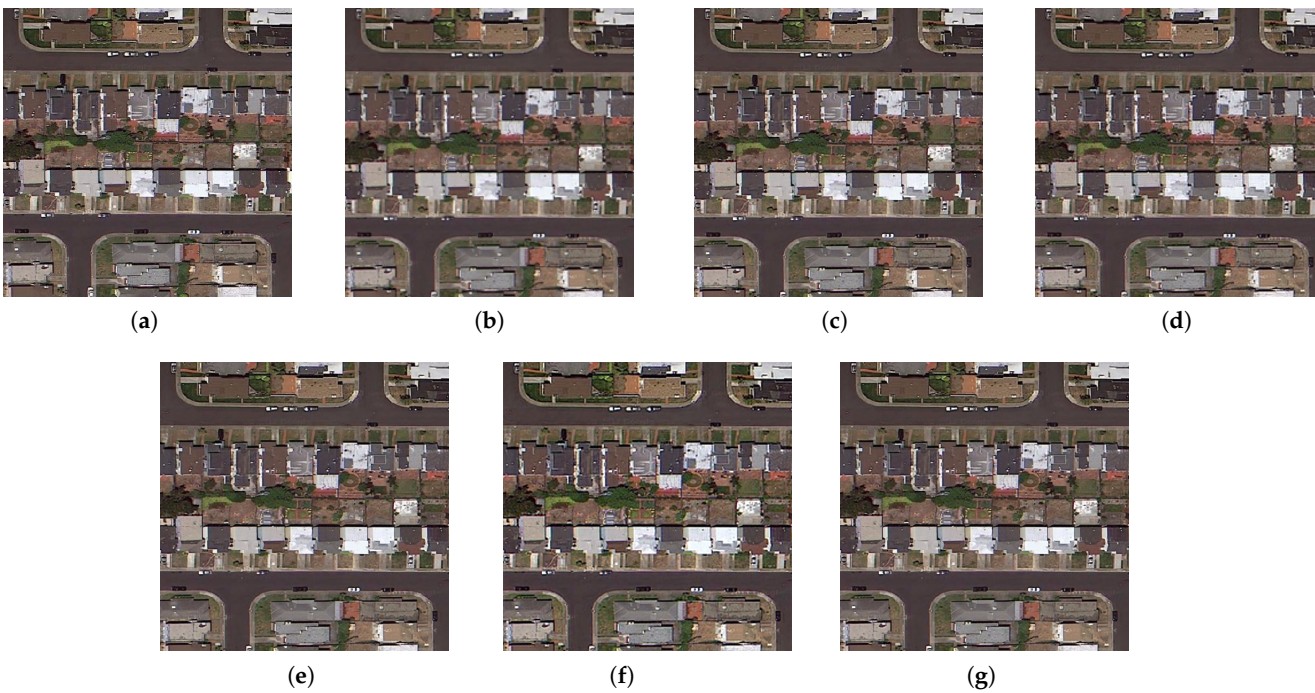

**Figure 13.** Comparison of visual results of different SR methods with ×2 super-resolution on the RSC11 [44] dataset (residential area scene). (**a**) HR. (**b**) Bicubic. (**c**) EDSR [68]. (**d**) RCAN [81]. (**e**) RDN [64]. (**f**) SAN [83]. (**g**) NLSN [88].

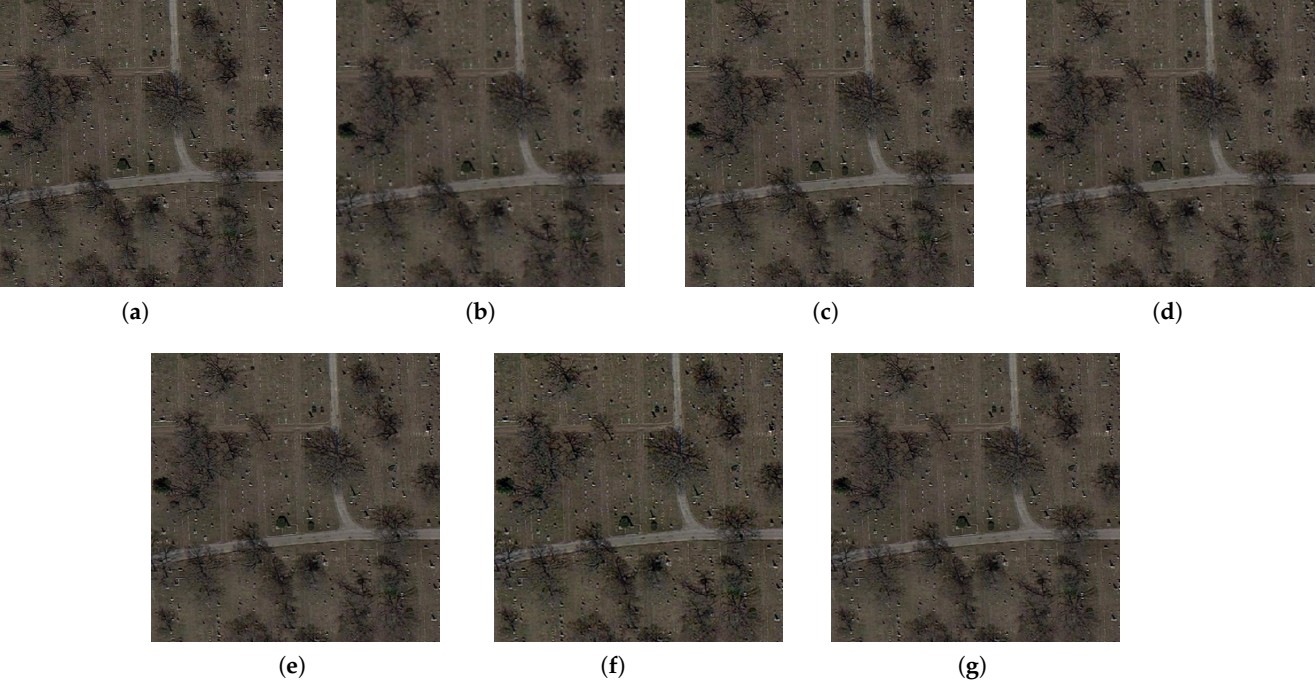

**Figure 14.** Comparison of visual results of different SR methods with ×2 super-resolution on the RSC11 [44] dataset (sparse forest scene). (**a**) HR. (**b**) Bicubic. (**c**) EDSR [68]. (**d**) RCAN [81]. (**e**) RDN [64]. (**f**) SAN [83]. (**g**) NLSN [88].

## 5. Remote Sensing Applications

Among the most critical factors for success in remote sensing applications, such as target detection and scene recognition, are high-resolution remote sensing images with

rich detail. Thus, methods of super-resolution that can be used for remote sensing have received more attention from researchers. The characteristics of remote sensing images have been addressed by many researchers in recent years by proposing super-resolution methods [138–142]. In this section, these methods are divided into two categories: supervised remote sensing image super-resolution and unsupervised remote sensing image super-resolution, and their characteristics are summarized.

### 5.1. Supervised Remote Sensing Image Super-Resolution

Most current remote sensing image super-resolution methods use supervised learning, i.e., LR–HR remote sensing image pairs are used to train models to learn the mapping from low-resolution remote sensing images to high-resolution remote sensing images.

In [143], a multiscale convolutional network MSCNN is proposed to accomplish remote sensing image feature extraction using convolutional kernels of different sizes to obtain richer, deeper features. Inspired by DBPN [91] and ResNet [63], Pan et al. proposed the residual dense inverse projection network (RDBPN) [134], which consists of projection units with dense residual connections added to obtain local and global residuals, while achieving feature reuse to provide more comprehensive features for large-scale remote sensing image super-resolution. Lei et al. [144] focused on remote sensing images containing more flat regions (i.e., more low-frequency features), and proposed coupled-discriminate GAN (CDGAN). In CDGAN, the discriminator receives inputs from both real HR images and SR images to enhance the network's ability to discriminate low-frequency regions of remote sensing images, and a coupled adversarial loss function is introduced to further optimize the network. In [145], a hybrid higher-order attention network (MHAN) is proposed, including two parts: a feature extraction network and feature refinement network. Among them, the higher-order attention mechanism (HOA) is used to reconstruct the high-frequency features of remote sensing images while introducing frequency awareness to make full use of the layered features. E-DBPN (Enhanced-DBPN) [144] is a generator network constructed based on DBPN. Enhanced residual channel attention module (ERCAM) is added to E-DBPN, which has the advantage of not only preserving the input image original features but also allowing the network to concentrate on the most significant portions of the remote sensing images, thus extracting features that are more helpful for super-resolution. Meanwhile, a sequential feature fusion module (SFFM) is proposed in E-DBPN to process the feature output from different projection units in a progressive manner. Usually, remote sensing images have a wide range of scene scales and large differences in object sizes in the scene. To address this characteristic of remote sensing images, Zhang et al. [146] proposed the multi-scale attention network (MSAN), which uses a multi-level activation feature fusion module (MAFB) to extract features at different scales and further fuse them. In addition, a scene adaptive training strategy is proposed to make the model better adapt to remote sensing images from different scenes. In [147], a deep recurrent network is proposed. First, the encoder extracts the remote sensing image features, a gating-based recurrent unit (GRU) is responsible for feature fusion, and finally the decoder outputs the super-resolution results. To reduce the computation and network parameters, Wang et al. [148] proposed a lightweight context transformation network (CTN) for remote sensing images. The context transformation layer (CTL) in this network is a lightweight convolutional layer, which can maintain the network performance while saving computational resources. In addition, the context conversion block (CTB) composed of CTL and the context enhancement module (CEM) jointly complete the extraction and enhancement of the contextual features of remote sensing images. Finally, the feature representation is processed by the context aggregation module to obtain the reconstruction results of remote sensing images. The U-shaped attention connectivity network (US-ACN) for the super-resolution of remote sensing images proposed by Jiang et al. [149] solves the problem of the performance degradation of previous super-resolution models on real images by learning the commonality of the internal features of remote sensing images. Meanwhile, a 3D attention module is designed to calculate 3D weights by learning channels

and spatial attention, which is more helpful for the learning of internal features. In addition, a U-shaped connection is added between the attention modules, which is more helpful for the learning of attention weights and the full utilization of contextual information. In [141], self-attention is used to improve the generative adversarial network and its texture enhancement function is used to solve the problems of edge blurring and artifacts in remote sensing images. The improved generator based on weight normalization mainly consists of dense residual blocks and a self-attentive mechanism for feature extraction, while stabilizing the training process to recover the edge details of remotely sensed images. In addition, a loss function is constructed by combining L1 parametric, perceptual, and texture losses, thus optimizing the network and removing remote sensing image artifacts. In [139], fuzzy kernel and noise are used to simulate the degradation patterns of real remote sensing images. The discriminator of Unet architecture is used to stabilize the training, while the residual balanced attention network (RBAN) is proposed to reconstruct the real texture of remote sensing images.

*5.2. Unsupervised Remote Sensing Image Super-Resolution*

Despite the fact that the super-resolution method with supervised learning has produced some results, there are still challenges associated with the pairing of LR–HR remote sensing images. On the one hand, the current remote sensing imaging technology and the influence of the external environment cannot meet the demand for high-resolution remote sensing images; on the other hand, the acquired high-resolution remote sensing images are processed with ideal degradation (such as double triple downsampling, Gaussian blur, etc.), and such degradation modes cannot approach the degradation of realistic low-resolution remote sensing images.

In [150], the generated random noise is first projected to the target resolution to ensure the reconstruction constraint on the LR input image, and the image is reconstructed using a generator network to obtain high-resolution remote sensing images by iterative iterations. In [151], a CycleGAN-based remote sensing super-resolution network is proposed. The training process uses the output of the degradation network as the input of the super-resolution network and the output of the super-resolution network as the input of the degradation network, so as to construct a cyclic loss function and thus improve the network performance. In [152], the unsupervised network UGAN is proposed. The network feeds low-resolution remote sensing images directly to the generator network and extracts features using convolutional kernels of different sizes to provide more information for the unsupervised super-resolution process. In [153], after training with a large amount of synthetic data, the most similar model to real degradation is developed, and then a loss function is derived from the difference between the original low-resolution image of the remote sensing network and the degraded image of the model.

## 6. Current Challenges and Future Directions

The models that have achieved excellent results in the field of image super-resolution in the past are presented in Section 3 and 4. The results of the application of these models on remotely sensed images show that they have driven the development of image super-resolution as well as remote sensing image processing techniques. The description of the methods for the super-resolution of remote sensing images in Section 5 also proves that this is a promising research topic. However, there are still many unresolved issues and challenges in the field of image super-resolution. Especially in the direction of the super-resolution of remote sensing images, on the one hand, remote sensing images, compared with natural images, are characterized by diverse application scenarios, a large number of targets, and complex types; on the other hand, external environments such as lighting and atmospheric conditions can affect the quality of remote sensing images. In this section, we will discuss these issues and introduce some popular and promising directions for future research. Remote sensing super-resolution can break through the limitations of technical level and environmental conditions, contributing to studies of resource development and

utilization, disaster prediction, etc. We believe that these directions will encourage excellent work to emerge on the topic of image super-resolution, and further explore the application of super-resolution methods to remote sensing images, contributing to the advancement of remote sensing.

### 6.1. Network Design

A proper network architecture design not only has high evaluation metrics but also enables efficient learning by reducing the running time and computational resources required, resulting in an excellent performance. Some promising future directions for network design are described below.

*(1) More Lightweight and Efficient Architecture.* Although the proposed deep network models have shown excellent results on several benchmark datasets and better results on various evaluation methods, the good performance of the models is determined by multiple factors, such as the number of model parameters and the resources required for computation, which determine whether the image super-resolution methods can be applied in realistic scenarios (e.g., smartphones and cameras, etc.). Therefore, it is necessary to develop lighter and more efficient image super-resolution network architectures to achieve higher research value. For example, compressing the model size using techniques such as network binarization and network quantization is a desirable approach. In the future, achieving a lightweight and efficient network architecture will be a popular trend in the field of image super-resolution. In the meantime, the application of the network architecture to the super-resolution of remote sensing images not only improves the reconstruction efficiency but also speeds up the corresponding remote sensing image processing.

*(2) Combination of Local and Global Information.* For image super-resolution tasks, the integrity of local information makes the image texture more realistic, and the integrity of global information makes the image content more contextually relevant. Especially for remote sensing images, the feature details are more severely corrupted compared with natural images. Therefore, the combination of local and global information will provide richer features for image super-resolution, which helps in the generation of complete high-resolution reconstructed images. In the practical application of remote sensing images, feature-rich high-resolution images play an invaluable role. For example, when using remote sensing technology for geological exploration, the observation and analysis of the spectral characteristics of remote sensing images enables the timely acquisition of the surface conditions for accurate judgment.

*(3) Combination of High-frequency and Low-frequency Information.* Usually, convolutional networks are good at extracting low-frequency information, and high-frequency information (such as image texture, edge details, etc.) is easily lost in the feature transfer process. Due to the limitation of the imaging principle of the sensor, the acquired remote sensing images also occasionally have the problem of blurred edges and artifacts. Improving the network structure by designing a frequency domain information filtering mechanism, combining it with a transformer, etc., to retain the high-frequency information in the image by as much as possible will help in the reconstruction of high-resolution images. When remote sensing technology is applied to vegetation monitoring, the complete spectral and textural features in remote sensing images will help improve the classification accuracy for vegetation.

*(4) Real-world Remote Sensing Image Super-resolution.* In the process of remote sensing image acquisition, realistic training samples of LR–HR remote sensing images are often not obtained due to atmospheric influence and imaging system limitations. On the one hand, the LR remote sensing images obtained by most methods using ideal degradation modes (such as double triple downsampling, Gaussian fuzzy kernel, and noise) still have some differences from the spatial, positional, and spectral information of the real remote sensing images. Therefore, the methods used to generate images that are closer to the real degraded remote sensing images are of important research value. On the other hand, unsupervised super-resolution methods can learn the degradation process of LR remote sensing images

and reconstruct them in super-resolution without pairwise training samples. Therefore, research on unsupervised remote sensing image super-resolution methods should receive more attention so as to cope with some real scenarios of remote sensing image super-resolution tasks.

*(5) Remote Sensing Image Super-resolution across Multiple Scales and Scenes.* The scenes of remote sensing images often involve multiple landscapes, and the target objects in the same scene vary greatly in size, which presents some challenges to the learning and adaptive ability of the model. Meanwhile, most current remote sensing image super-resolution methods use ×2, ×3, and ×4 scale factors. As a consequence, the model should be trained to learn how to map relationships between LR–HR remote sensing images from multiple scenes. For the characteristics of target objects in remote sensing images, more attention should be paid to the research of super-resolution methods with ×8 and larger scale factors, so as to provide more useful information for remote sensing image processing tasks.

### 6.2. Learning Strategies

In addition to the network architecture design, a reasonable deep learning strategy is also an important factor in determining the network performance. Some promising learning strategy design solutions are presented here.

*(1) Loss Function.* Most of the previous network models choose MSE loss or L2 loss or use a weighted combination of loss functions. Most suitable loss functions for image super-resolution tasks are still to be investigated. Although some new loss functions have been proposed from other perspectives, such as perceptual loss, content loss, and texture loss, they have yet to produce satisfactory results regarding their applications in image super-resolution tasks. Therefore, it is necessary to further explore the balance between image super-resolution accuracy and perceptual quality to find more accurate loss functions.

*(2) Batch Normalization.* Batch normalization speeds up model training and has been widely used in various computer vision tasks. Although it solves the gradient disappearance problem, it is unsatisfactory for image super-resolution in some studies. Therefore, the normalization techniques suitable for super-resolution tasks need further research.

### 6.3. Evaluation Methods

Image quality evaluation, as an essential procedure in the process of image super-resolution based on deep learning, also faces certain challenges. How to propose an evaluation metric with simple implementation and accurate results still needs to be continuously explored. Some promising development directions to solve the current problem are presented below.

*(1) More Precise Metrics.* PSNR and SSIM, as currently popular evaluation metrics, also have some drawbacks. Although PSNR is a simple algorithm that can be implemented quickly, because it is a purely objective evaluation method, the calculated results sometimes differ greatly from those obtained by human vision. SSIM measures the quality of reconstructed images in terms of brightness, contrast, and structure. However, there are some limitations on the evaluation objects, and for images that have undergone non-structural distortion (e.g., displacement, rotation, etc.), SSIM cannot evaluate them properly. Therefore, it is necessary to propose a more accurate image evaluation index.

*(2) More Diverse Metrics.* As image super-resolution technology continues to advance, it is used in more fields. In this case, it is inaccurate to use only mainstream evaluation metrics such as PSNR or SSIM to evaluate reconstruction results. For example, reconstructed images applied in the medical field tend to focus more on the recovery of detailed areas, and it is necessary to refer to evaluation criteria that focus on the high-frequency information of the image. MOS, as a subjective evaluation method, evaluates the results in a manner that is closer to the visual perception of the human eye, but in practice, it is difficult to implement this method because it requires a large number of people to participate. There is a need to propose more targeted evaluation indices for certain characteristics of remote sensing images in particular. The spatial resolution and spectral resolution of remote sensing

images play a vital role in practical applications, such as weather forecasting, forestry, and geological surveying, etc. Thus, to evaluate the quality of reconstructed remote sensing images, one should consider whether the reconstruction results can optimize a particular property of these images. In general, the diversification of image evaluation metrics is also a popular development direction.

## 7. Conclusions

This paper provides a comprehensive summary of deep-learning-based image super-resolution methods, including common datasets, image quality evaluation methods, model reconstruction efficiency, deep learning strategies, and some techniques to optimize network metrics. In addition, the applications of image super-resolution methods in remote sensing images are comprehensively presented. Finally, although the research on image super-resolution methods, especially for remote sensing image super-resolution reconstruction, has made great progress in recent years, significant challenges remain, such as low model inference efficiency, the unsatisfactory reconstruction of real-world images, and a single approach to measuring the quality of images. Thus, we point out some promising development directions, such as more lightweight and effective model design strategies, remote sensing image super-resolution methods that are more adaptable to realistic scenes, and more accurate and diversified image evaluation metrics. We believe this review can help researchers to gain a deeper understanding of image super-resolution techniques and the application of super-resolution methods in the field of remote sensing image processing, thus promoting progress and development.

**Author Contributions:** Conceptualization, X.W., Y.S. and W.Y.; software, J.Y. and J.G.; investigation, J.Y., J.X. and J.L.; formal analysis, Q.C.; writing—original draft preparation, J.Y., X.W. and H.M.; writing—review and editing, Q.C., W.Y. and Y.S.; supervision, J.Z., J.X., W.Y. and J.L.; funding acquisition, X.W., J.X., J.Z., Q.C. and H.M. All authors have read and agreed to the published version of the manuscript.

**Funding:** This research was funded by the Natural Science Foundation of Shandong Province (ZR2020QF108, ZR2022QF037, ZR2020MF148, ZR2020QF031, ZR2020QF046, ZR2022MF238), and the National Natural Science Foundation of China (62272405, 62072391, 62066013, 62172351, 62102338, 62273290, 62103350), and in part by the China Postdoctoral Science Foundation under Grant 2021M693078, and Shaanxi Key R & D Program (2021GY-290), and the Youth Innovation Science and Technology Support Program of Shandong Province under Grant 2021KJ080, Yantai Science and Technology Innovation Development Plan Project under Grant 2021YT06000645, the Open Foundation of State key Laboratory of Networking and Switching Technology (Beijing University of Posts and Telecommunications) under Grant SKLNST-2022-1-12.

**Institutional Review Board Statement:** Not applicable.

**Informed Consent Statement:** Not applicable.

**Data Availability Statement:** The datasets are available on Github at https://github.com/Leilei111 11/DOWNLOADLINK, accessed on 28 September 2022.

**Acknowledgments:** We would like to thank the anonymous reviewers for their supportive comments to improve our manuscript.

**Conflicts of Interest:** The authors declare no conflict of interest.

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
