# Peer review of "A Review of Image Super-Resolution Approaches Based on Deep Learning and Applications in Remote Sensing"

_remotesensing, doi:10.3390/rs14215423_

Round 1

Reviewer 1 Report

This paper presented a comprehensive summary of image super-resolution methods based on deep learning, and also describes the application of image super-resolution methods to remote sensing images. This is an interesting research paper. There are some suggestions for revision.

1. In line 4, why use image super-resolution methods used in remote sensing image reconstruction? Consider adding some advantages of these methods.

2. The paper mainly describes the remote sensing super-resolution image reconstruction, but in Figure3 the LR and HR are others but remote sensing images. Consider changing images in this Figure.

3. There are certain differences between ordinary images and remotely sensed images, and relevant background and work on remote sensing image processing should be discussed in Section 2.

4. Section 4 should give a more intuitive comparison and analysis for the comparison of various models. It is suggested to add the experimental results and analysis of the visualization of super-resolution methods on remote sensing images.

5. Section 6 proposes to add details about the evaluation metrics of remote sensing images, such as spatial resolution, spectral resolution, etc.

6. Line 828, the word “problems” in the conclusion may be replaced by a more appropriate term.

7. Other methods used for super-resolution of remotely sensed images should be presented in Table 2, and conclusions should be drawn from the comparative analysis.

Author Response

Reviewer #1:

  1. In line 4, why use image super-resolution methods used in remote sensing image reconstruction? Consider adding some advantages of these methods.

Answer: In line 6, we have reanalyzed and described the advantages of image super-resolution methods in remote sensing image reconstruction.

  1. The paper mainly describes the remote sensing super-resolution image reconstruction, but in Figure3 the LR and HR are others but remote sensing images. Consider changing images in this Figure.

Answer: We have replaced the LR and HR images in Figure 3 with remotely sensed images.

  1. There are certain differences between ordinary images and remotely sensed images, and relevant background and work on remote sensing image processing should be discussed in Section 2.

Answer: We have added background information and related work on remote sensing images in Section 2, shown in line 121.

  1. Section 4 should give a more intuitive comparison and analysis for the comparison of various models. It is suggested to add the experimental results and analysis of the visualization of super-resolution methods on remote sensing images.

Answer: We have added experimental results and analysis of the visualization of the classical model on remote sensing images in Section 4.

  1. Section 6 proposes to add details about the evaluation metrics of remote sensing images, such as spatial resolution, spectral resolution, etc.

Answer: We have described the details of the remote sensing image evaluation metrics in Section 6, shown in line 908.

  1. Line 828, the word “problems” in the conclusion may be replaced by a more appropriate term.

Answer: On line 921, we have replaced the description with a more appropriate one.

  1. Other methods used for super-resolution of remotely sensed images should be presented in Table 2, and conclusions should be drawn from the comparative analysis.

Answer: To provide a more comprehensive overview of other methods used for super-resolution of remote sensing images, we added Table 3 to enumerate the methods and analyze them.

Reviewer 2 Report

This paper mainly provided a comprehensive overview and analysis of deep learning-based image super-resolution methods. The organization of the manuscript is good to some extent, however, there are still some problems with the organization of those sub-subsections. The literature review is quite sufficient and the technical issues are almost correct. This review has the following concerns:

(1) The English should be polished by a native English professor to correct some grammar and spelling mistakes.

(2) In figure 2, it is suggested to add some representative methods for each sub-topics.

(3) The citation of the channel attention used in Figure 7 should be addressed.

(4)For those sub-subsections, the organization should be modified. 

(5) For Section 4 "analysis and comparisons of various models", the content should be expanded.

(6) Analysis and summary of challenges should be in-depth and constructive

Author Response

Reviewer #2:

  1. The English should be polished by a native English professor to correct some grammar and spelling mistakes.

Answer: We used the MDPI english editing services to correct the grammar and spelling mistakes. The paper has undergone English language editing by MDPI. It is confirmed that the text has been checked for correct use of grammar and common technical terms, and edited to a level suitable for reporting research in a scholarly journal.

  1. In figure 2, it is suggested to add some representative methods for each sub-topics.

Answer: We have added representative methods for each of the sub-topics in Figure 2.

  1. The citation of the channel attention used in Figure 7 should be addressed.

Answer: We have modified the citation of the channel attention used in Figure 7.

  1. For those sub-subsections, the organization should be modified.

Answer: We have modified the organization of the subsections in Section 6.

  1. For Section 4 "analysis and comparisons of various models", the content should be expanded.

Answer: We have listed and analyzed the classical remote sensing image super-resolution methods (as shown in Table 3) in Section 4 (as shown in line 656), and added the visualization results (as shown in Figures 9,10,11,12,13,14) and analysis of the classical image super-resolution methods on the remote sensing datasets WHU-RS19 and RSC11.

  1. Analysis and summary of challenges should be in-depth and constructive.

Answer: We have revised the analysis and summary of the challenges to be more in-depth and constructive as shown in lines 798, 804, 823, 834, and 846.

Reviewer 3 Report

The authors presented a review of the superresolution of the remote-sensing images. As for me, the paper is valuable for the research, but there are some limitations included that should be considered, which are listed as follows: 

1- The topic " A Review of Image Superresolution and Remote Sensing Applications" is unclear. Please correct the topic.

2-Currently we have excellent papers about superresolution and its applications in remote sensing; why do we need such a review? For example, please see the following papers in this way: 

https://doi.org/10.1117/1.OE.60.10.100901

https://doi.org/10.1016/j.earscirev.2022.104110

https://doi.org/10.1109/ICDSP.2015.7251858

3- Many researchers have developed superresolution differently, not only Deep learning (DL) methods. Please refer to conventional methods as well as DL methods. 

4- The considered SR method lacks any visual comparison. Please show at least two datasets' visual results of the considered method. 

Author Response

Reviewer #3:

  1. The topic " A Review of Image Super-resolution and Remote Sensing Applications" is unclear. Please correct the topic.

Answer: We have corrected the title to "A Review of Image Super-resolution Approaches Based on Deep Learning and Applications in Remote Sensing".

  1. Currently we have excellent papers about super-resolution and its applications in remote sensing; why do we need such a review? For example, please see the following papers in this way:

https://doi.org/10.1117/1.OE.60.10.100901

https://doi.org/10.1016/j.earscirev.2022.104110

https://doi.org/10.1109/ICDSP.2015.7251858

Answer: We believe that the above three papers focus more on the summary of the remote sensing image super-resolution. In our paper, we mainly review the deep learning-based image super-resolution methods, which are mostly used for super-resolution of natural images, and we explore the application of these methods on remote sensing images. Specifically, we select representative SR methods for experiments on remote sensing image datasets, discuss and analyze the visualization results of the experiments, and finally propose some existing problems and development directions. We hope that this paper will help to promote the progress of image super-resolution technology and bring more value to the field of remote sensing.

  1. Many researchers have developed super-resolution differently, not only Deep learning (DL) methods. Please refer to conventional methods as well as DL methods.

Answer: In the introduction section, we added the introduction and analysis of traditional methods as shown in line 37, 45 and 48. Since we corrected the title, this paper focuses on reviewing and analyzing deep learning based SR methods, so there is no more review and analysis of traditional methods.

  1. The considered SR method lacks any visual comparison. Please show at least two datasets' visual results of the considered method.

Answer: In Section 4.2, we show the visualization results of the SR method on the square, parking lot, and forest scenes in the WHU-RS19 dataset (shown in Figure 9, 10, and 11) and the port, residential area, and sparse forest scenes in the RSC11 dataset (shown in Figure 12, 13, and 14). We further compare and analyze the relevant algorithms with the visualization results.

Round 2

Reviewer 1 Report

All reviewer comments have been addressed and addressed satisfactorily. The reviewer would commend the authors for their time and effort in the work aiming to deliver the paper with academic merit. The reviewer would thus, like to recommend "accept" to the paper.

Reviewer 2 Report

All my concerns have been satisfactorily modified, there are no more comments.

Reviewer 3 Report

The authors have addressed all of my concerns.